



# A control-oriented dynamic wind farm model: WFSim

Sjoerd Boersma[1], Bart Doekemeijer[1], Mehdi Vali[2], Johan Meyers[3], and Jan-Willem van Wingerden[1]

[1]Delft University of Technology, Delft Center for Systems and Control, Mekelweg 2, 2628 CC, Delft, Holland.
[2]Wind Energy System Research Group, ForWind, Küpkersweg 70, 26129 Oldenburg, Germany.
[3]KU Leuven, Department of Mechanical Engineering, Celestijnenlaan 300A, B3001 Leuven, Belgium.

*Correspondence to:* Sjoerd Boersma (s.boersma@tudelft.nl).



**Abstract.**

Wind turbines are often sited together in wind farms as it is economically advantageous. Controlling the flow within wind farms to reduce the fatigue loads and provide grid facilities such as the delivery of a demanded power is a challenging control problem due to the underlying time–varying nonlinear wake dynamics. It is therefore important to use the closed–loop control paradigm since it can partially account for model uncertainty and, in addition, it can deal with unknown disturbances. State–of–the–art closed–loop dynamic wind farm controllers are based on computationally expensive wind farm models, which make these methods suitable for analysis though unsuitable for online control. The latter is important, because it allows for model adaptation to the time–varying atmospheric conditions using SCADA measurements. As a consequence, more reliable control settings can be evaluated.

In this paper, a dynamic wind farm model suitable for online wind farm control will be presented. The derivation of the control–oriented dynamic wind farm model starts with the three-dimensional Navier–Stokes equations. Then, terms involving the vertical dimension will be estimated in order to partially compensate for neglecting the vertical dimension or neglected such that a 2D–like dynamic wind farm model will be obtained. Sparsity of and structure in the system matrices make this model relatively computational inexpensive hence suitable for online closed–loop controller synthesis including model parameter updates. Flow and power data evaluated with the wind farm model presented in this work will be validated with high fidelity flow data.

**Table 1.** Nomenclature.

| | | | |
|---|---|---|---|
| $L_x \times L_y$, | domain size wind farm | $D$ | turbine rotor diameter |
| $N_x \times N_y$, | number of cells | $\Delta x \times \Delta y$ | cell size |
| $T_n$, | turbine $n$ | $\aleph$, | number of turbines |
| $U_n$, | hub-height flow velocity at the rotor | $u_b, v_b$, | inflow conditions |
| $U^c$, | flow centreline velocity | $U_\infty$, | upstream flow velocity |
| $C_T, C_P$, | thrust force and power coefficient | $\boldsymbol{f}$, | wind turbine force |
| $l_u$, | turbulence model parameter | $\boldsymbol{\tau}_H$, | 2D stress tensor |
| $\Delta t$, | sample period | $k$, | time index |
| $q_k = \begin{pmatrix} u_k^T & v_k^T & p_k^T \end{pmatrix}^T$, | state vector with longitudinal and lateral flow velocities and pressure | $n_q$, | number of states |
| $w_k = \begin{pmatrix} \nu_k^T & \gamma_k^T \end{pmatrix}^T$, | control variables | $z_k$, | measurement vector |



# 1 Introduction

Optimizing the control of wind turbines in a farm is challenging due to the aerodynamic interactions among turbines. These interactions come from the fact that downwind turbines are often operating in the wakes of upwind ones (Barthelmie et al., 2009). Two important wake characteristics are: 1) a flow velocity deficit and 2) an increase in turbulence intensity. The former reduces

power production of the farm while the latter leads to a higher dynamic loading on downstream turbines, but also induces wake recovery. Individual turbine control variables can influence the wake's flow velocity, turbulence intensity and also location. Hence, by changing the control variables of individual turbines, power production of and loading on these controlled turbines and its downwind turbines can be manipulated. Wind farm control aims to find control variables under changing atmospheric conditions such that demanded power production and/or a minimization of the loading can be guaranteed, improving the cost

and quality of wind energy. A survey on wind farm control can be found in, *e.g.*, (Knudsen et al., 2015; Boersma et al., 2017). In the latter, a clear distinction is made between model based and model free control algorithms. This manuscript is focussed on the former where it is assumed that controllers are based on a mathematical model of the system. Consequently, the controller performance depends highly on the model quality. Modelling is therefore a crucial step towards successful implementation of model based wind farm control.

Overviews on wind farm models can be found in (Crespo et al., 1999; Vermeer et al., 2003; Sanderse, 2009; Sanderse et al., 2011; Annoni et al., 2014; Göçmen et al., 2016; Boersma et al., 2017). The spectrum of these models ranges from low fidelity to high fidelity. The latter tries to capture relatively precise wind farm flow and turbine dynamics, while the former tries to capture only the dominant characteristics (dynamic or static) in a wind farm. Examples of high fidelity wind farm models are Simulator fOr Wind Farm Applications (SOWFA) (Churchfield et al., 2012), UTD Wind Farm (UTDWF) (Martinez-Tossas et al.,

2014), SP–Wind (Meyers and Meneveau, 2010) and PArallelized LES Model (PALM) (Maronga et al., 2015). These three dimensional (3D) high fidelity wind farm models can easily have $10^6$ or more states. The resulting computation time can be in order of days or weeks using distributed computation for simulation times less than the computation time. In other words, the computation time needed for LES is in general more than the total time that is simulated. Clearly, these types of models are not applicable for online model–based control. Rather, these models serve as analysis/validation tools.

One way to reduce the high complexity of wake modeling is by using two–dimensional (2D) heuristic models that only capture specific wake and turbine characteristics in a wind farm in the horizontal plane at hub height. These type of models are found on the low fidelity side of the spectrum. Most of these wake models exclusively estimate a steady state situation for given atmospheric conditions. Examples of static models are the Frandsen model (Frandsen et al., 2006) and the Jensen Park model (Jensen, 1983; Katic et al., 1986). One extension of the Jensen model resulted in the parametric model called FLOw

Redirection and Induction in Steady–state (FLORIS) (Gebraad et al., 2014b). Two examples of low fidelity dynamic models are SimWindFarm (Grunnet et al., 2010) and the model used in (Shapiro et al., 2017a), where relatively simple approximations of the flow deficit are computed using heuristic expressions.

Medium fidelity models can be found in the middle of the spectrum as they trade off the accuracy of high fidelity models, with the computational complexity of low fidelity models. These are in general based on simplified versions of the Navier–



Stokes equations. For example, in the 2D Ainslie (Ainslie, 1988) model and the 2D Dynamic Wake Meandering (DWM) model (Larsen et al., 2007), assumptions are made regarding the thin shear layer such that the Navier–Stokes equations can be approximated using less computational effort. The authors in (Trabucchi et al., 2016) present a model, which is also based on the thin shear layer approximation, but according to the authors applicable for non–axisymmetric wind turbine wakes.

WakeFarm (also referred to as Farmflow) simulates the wind turbine wakes by solving the steady parabolized Navier–Stokes equations in three dimensions (Crespo et al., 1988; Özdemir et al., 2013). In (Annoni and Seiler, 2015), time averaging is applied on the Navier–Stokes equations resulting in the Reynolds Averaged Navier–Stokes (RANS) equations. The number of states is then reduced by employing a state reduction technique.

Also considered as medium fidelity models are the ones presented in (Boersma et al., 2016b; Soleimanzadeh et al., 2014).
These wind farm models are based on the discretized 2D Navier–Stokes equations. However, these models do not contain a turbulence model that allows for wake recovery. In addition, these 2D models do not take any neglected 3D effects into account and no yaw actuation of the individual turbines is included.

In this paper, a model will be presented that can be considered as a building block for the closed-loop control framework as illustrated in Fig. 1.

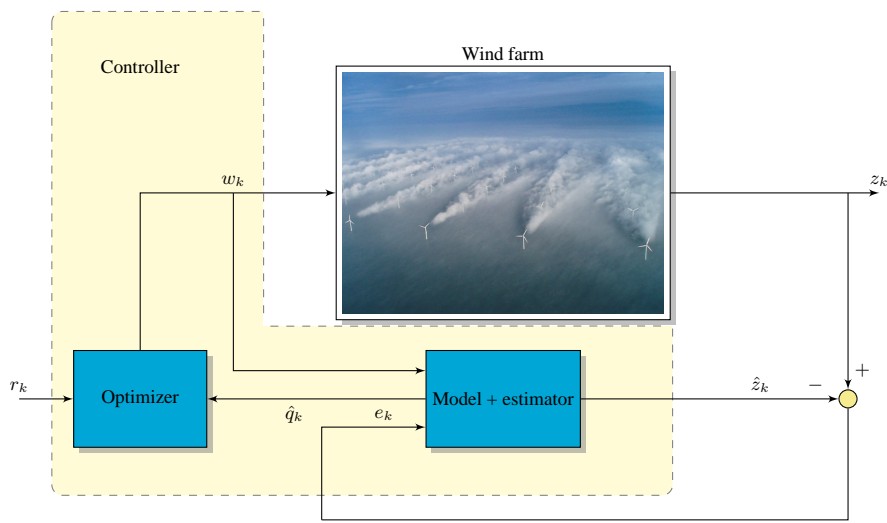

**Figure 1.** General dynamic closed–loop dynamic control framework with measurements $z_k$ and its estimation $\hat{z}_k$ and state estimation $\hat{q}_k$. The signals $r_k$ and $w_k$ are a reference and control signal, respectively. In this paper we present a dynamic model that is compatible with this framework.

In current practice, signals such as power can be measured from a wind farm, but current research is also focussing on estimating wake characteristics using a LIDAR device (Raach et al., 2017). These and other wind farm measurements are called SCADA data and can be used by an estimator that is able to adapt the model parameters to current atmospheric conditions and/or estimate the full state space, *e.g.*, all the flow velocities at hub height in the farm. The work presented

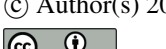



in (Doekemeijer et al., 2016) illustrates the latter and employs the dynamic wind farm model presented in this paper. Subsequently, the estimation can then be used to compute optimal control variables using a model predictive controller. The work presented in (Vali et al., 2016) illustrates the application of such a model predictive wind farm controller using the dynamic model presented in this work.

The online closed–loop control paradigm as depicted in Fig. 1 demands for a control–oriented dynamic wind farm model that will be presented in this paper. Characteristics of such control–oriented model are, *i.a.*:

  1. Low computational cost such that online model update, state estimation and control signal evaluation is possible.

  2. Dynamic such that it can deal with varying atmospheric conditions within relatively small time scales.

The dynamic control–oriented wind farm model presented in this paper, referred to as WindFarmSimulator (WFSim), is appli-
cable in the framework discussed above and satisfies the above two points. It is based on corrected 2D Navier–Stokes equations and contains a heuristic turbulence model. The Navier–Stokes equations are modified in order to partially correct for the neglected vertical dimension. Each turbine is modelled using the actuator disk model (ADM) and features yaw and axial induction actuation. An important model feature is the exploitation of the sparse system matrices, leading to computational efficiency. WFSim will be compared to high fidelity flow data and used in a practical control application.

The remainder of this paper is organized as follows. In Section 2, the mathematical background of the medium fidelity wind farm model will be explained including a discussion on the rotor and turbulence model. This section ends with an analysis regarding the wind farm model its computation time. In Section 3, WFSim will be validated using high fidelity simulation data. This paper is concluded in Section 4.



## 2 Formulation of a dynamic control–oriented wind farm model

In the current section, a simplified wind–farm model is formulated that is sufficiently fast for online control, but retains some of the elemental features of three–dimensional turbulent flows. In order for the model to be fast, we envisage a 2D–like model, but adapted to account for three–dimensional flow relaxation. We will dub the resulting model WFSim (WindFarmSimulator).

As starting point we use the standard incompressible three–dimensional filtered Navier–Stokes equations, as used in large–eddy simulations (LES), *i.e.*

$$\frac{\partial \widetilde{\boldsymbol{v}}}{\partial t} + (\widetilde{\boldsymbol{v}} \cdot \nabla)\widetilde{\boldsymbol{v}} + \nabla \cdot \boldsymbol{\tau}_M + \frac{1}{\rho}\nabla \widetilde{p} - \boldsymbol{f} = 0, \qquad \text{momentum equations,}$$

$$\nabla \cdot \widetilde{\boldsymbol{v}} = 0, \qquad\qquad\qquad\qquad \text{continuity equation.} \tag{1}$$

The velocity field $\widetilde{\boldsymbol{v}} = (\widetilde{v}_1, \widetilde{v}_2, \widetilde{v}_3)^T$ and pressure field $\widetilde{p}$ represent filtered variables, $\nabla = (\partial/\partial x, \partial/\partial y, \partial/\partial z)^T$, the air density $\rho$, which is assumed to be constant, and $\boldsymbol{\tau}_M$ represents the subgrid scale model, that will be defined in §2.1. As common in LES

of high–Reynolds number atmospheric simulations with grid resolutions in the meter range, direct effects of viscous stresses on the filtered fields are negligible, so that these terms are left out. Finally, the term $\boldsymbol{f}$ represents the effect of turbines on the flow, as further detailed in §2.2.

Although LES filters are usually implicitly tied to the LES grid and filter length scale in the subgrid scale model, we presume here that $\widetilde{\boldsymbol{v}}$ corresponds to a top–hat filtered velocity field, with filter width $D$, where $D$ is the turbine diameter. Thus,

$$\widetilde{\boldsymbol{v}}(x,y,z) = \frac{1}{D^3} \int\limits_{z-D/2}^{z+D/2} \int\limits_{y-D/2}^{y+D/2} \int\limits_{x-D/2}^{x+D/2} \boldsymbol{v}(x',y',z')\,\mathrm{d}x'\mathrm{d}y'\mathrm{d}z'. \tag{2}$$

From a wind farm simulation perspective, we are mainly interested at the flow velocity field at hub height $z_h$, *i.e.*, $\widetilde{\boldsymbol{v}}(x,y,z_h)$. Moreover, to evaluate turbine forces and power, it suffices to know the velocity at turbine locations $\boldsymbol{t}_n = (x_n, y_n)^T$ (with $n = 1 \cdots \aleph$ and $\aleph$ the number of turbines in the farm), since $\widetilde{\boldsymbol{v}}(x_n, y_n, z_h)$ is a reasonable representation of the turbine disk-averaged velocity.

Therefore, we focus on formulating a 2D–like set of equations for $\widetilde{\boldsymbol{v}}(x,y,z_h)$. To this end, define:

$$\boldsymbol{u} = \begin{pmatrix} \widetilde{v}_1(x,y,z_h) & \widetilde{v}_2(x,y,z_h) \end{pmatrix}^T, \tag{3}$$

$$= \begin{pmatrix} u & v \end{pmatrix}^T, \tag{4}$$

and $w = \widetilde{v}_3(x,y,z_h)$ and $p = \widetilde{p}(x,y,z_h)/\rho$. Moreover, we assume that $w \approx 0$, so that the LES equations given in (1) can be reformulated in terms of $\boldsymbol{u}$ as

$$\frac{\partial \boldsymbol{u}}{\partial t} + (\boldsymbol{u} \cdot \nabla_H)\boldsymbol{u} + \nabla_H \cdot \boldsymbol{\tau}_H + \nabla_H p - \boldsymbol{f} = -\frac{\partial(uw + \tau_{M,13})}{\partial z}\boldsymbol{e}_1 - \frac{\partial(vw + \tau_{M,23})}{\partial z}\boldsymbol{e}_2, \tag{5}$$

$$\nabla_H \cdot \boldsymbol{u} = -\frac{\partial w}{\partial z}, \tag{6}$$

with $\nabla_H = (\partial/\partial x, \partial/\partial y)^T$, $\boldsymbol{\tau}_H$ a 2D tensor containing the horizontal components of the subgrid stresses $\boldsymbol{\tau}_M$, and $\boldsymbol{e}_1$ and $\boldsymbol{e}_2$ the unit vectors in $x$ and $y$ direction, respectively.

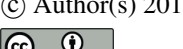


Finally, we further simplify above equations using two additional assumptions. First of all, we presume $\frac{\partial w}{\partial z} \approx \frac{\partial v}{\partial y}$. For a single turbine, this essentially corresponds to an assumption of axisymmetry. Secondly, we simply neglect the right–hand side of (5). Though this is a rather crude assumption, the rationale is that the modelling term $\boldsymbol{\tau}_H$ will suffice in the context of a control model, where model coefficients can be updated online based on feedback (see also the discussion in §2.1). Hence our

5  final 2D–like model corresponds to:

$$\frac{\partial \boldsymbol{u}}{\partial t} + (\boldsymbol{u} \cdot \nabla_H)\boldsymbol{u} + \nabla_H \cdot \boldsymbol{\tau}_H + \nabla_H p - \boldsymbol{f} = 0, \tag{7}$$

$$\nabla_H \cdot \boldsymbol{u} = -\frac{\partial v}{\partial y}. \tag{8}$$

We emphasize here that above model is not a classical 2D model due to the difference in formulation of the continuity equation. In contrast to a standard 2D model, this allows for flow relaxation in the third direction when, *e.g.*, encountering

10  slow down by a wind turbine. This can be seen in Fig. 2, where simulation results are shown obtained with the above model, a standard 2D dynamic wind farm model and LES. The simulation case itself will be discussed more detailed in §3.2.1. Here we depict the normalised flow deficit in the wake at $5D$ downstream of the turbine along the cross–stream axis. The figure

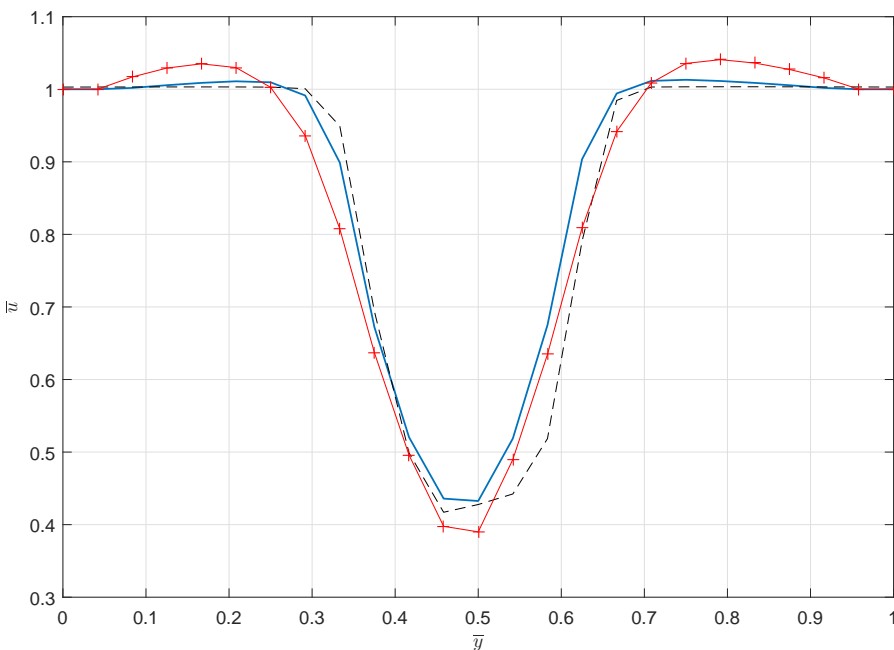

**Figure 2.** Results of two–turbine simulations. Normalised time–averaged wake deficit at hub height $5D$ downwind the downwind turbine using standard 2D Navier–Stokes equations (red crossed), our model with adapted continuity equation (blue), and LES data (black dashed).

illustrates that the standard 2D Navier–Stokes equations lead to a significant speed up at the wake edges. This is a result from conservation of mass in two dimensions and the flow deceleration in front of the turbine, pushing part of the air around the





turbine. In the WFSim model, this speed up is smaller, as mass can also flow around the turbine in the third dimension. In Fig. 2 it can be seen that LES data is better estimated when imposing flow relaxation in the third dimension. Finally, note that partially modelling the missing vertical dimension as proposed above is novel with respect to the work presented in (Boersma et al., 2016a).

This section is further organized as follows. First, in §2.1, the subgrid–scale model will be introduced. Then, in §2.2, the turbine model will be explained. The discretization of the equations is presented in §2.4, and boundary and initial conditions are discussed in §2.5.

## 2.1   Turbulence model

In the literature, many subgrid–scale models are documented, and to date, model accuracy remains a challenge in LES research
(see *e.g.*, (Sagaut, 2006)). However, in the current manuscript, an important factor in the selection of a model is simplicity and computational efficiency, rather than accuracy. In fact, in contrast to conventional modelling, in a control model completeness of the turbulence model is not a major issue, since unknown model coefficients can be calibrated online using measurements and feedback (Shapiro et al., 2017b), thus also controlling the overall error. Therefore, in this work we fall back to one of the simplest and first known turbulence models, Prandtl's mixing length model.

We formulate the stress tensor $\boldsymbol{\tau}_H$ using an eddy–viscosity assumption, *i.e.*

$$\boldsymbol{\tau}_H = -\nu_t \boldsymbol{S}, \tag{9}$$

with $\boldsymbol{S} = \frac{1}{2}(\nabla_H \boldsymbol{u} + (\nabla_H \boldsymbol{u})^T)$ the 2D rate–of–strain tensor, and $\nu_t$ the eddy viscosity. The latter is further modelled as (Prandtl, 1925):

$$\nu_t = l_u(x,y)^2 \left| \frac{\partial u}{\partial y} \right|, \tag{10}$$

where $l_u(x,y)$ is the mixing length. It could be interesting to define the mixing length for each position in the wind farm separately, but this will lead to too many tuning variables. Moreover, in (Iungo et al., 2015), the authors illustrate that in a turbine's near wake the mixing length is roughly invariant for different downstream locations, but in the far wake, the mixing length increases linearly with downstream distance. We use this to formulate a simple heuristic parametrization for the mixing length model so that the number of decision variables will be reduced drastically. From now on we assume that the wind is
coming from the east, but can have a direction defined by $\varphi$. Then, the wind farm will be divided in segments as illustrated in Fig. 3.

Each segments has its own $(x'_n, y'_n)$ coordinate system located in the global $(x,y)$ coordinate system. Now we propose the following mixing length parametrization:

$$l_u(x,y) = \begin{cases} G(x'_n, y'_n) * l_u^n(x'_n, y'_n), & \text{if } x \in \mathcal{X} \text{ and } y \in \mathcal{Y}. \\ 0, & \text{otherwise,} \end{cases} \tag{11}$$

with $G(x,y)$ a (smoothing) pillbox filter with radius 3, $*$ the 2D spatial convolution operator and $\mathcal{X} = \{x : x'_n \leq x \leq x'_n + \cos(\varphi)d\}$ and $\mathcal{Y} = \{y : y'_n - \frac{D}{2} + \sin(\varphi)x'_n \leq y \leq y'_n + \frac{D}{2} + \sin(\varphi)x'_n\}$ and $\varphi$ defined as the mean wind direction (see Fig. 3),

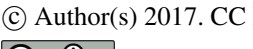



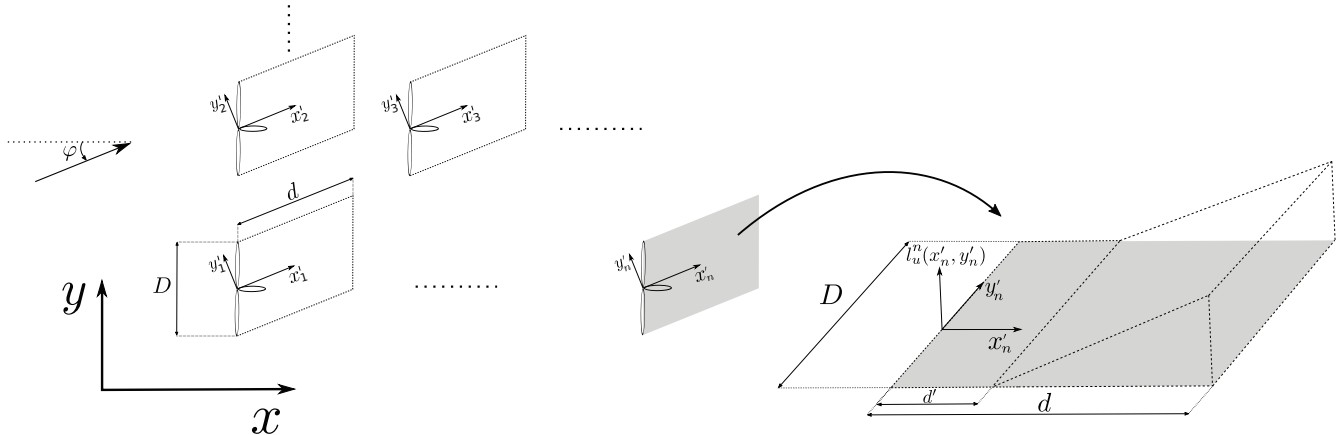

**Figure 3.** Schematic illustration of the mixing length.

which we in this work bound by $|\varphi| \leq 45°$. In addition we constraint $d$ by $\cos(\varphi)d \leq |x_q - x_n|$ with $x_n$ a turbines x–coordinate and $x_q$ its downwind turbines x–coordinate. We can see $l_u^n(x_n', y_n')$ as the local mixing length that belongs to turbine $n$ and denote it as:

$$l_u^n(x_n', y_n') = \begin{cases} (x_n' - d')l_s, & \text{if } x_n' \in \mathcal{X}_n' \text{ and } y_n' \in \mathcal{Y}_n'. \\ 0, & \text{otherwise.} \end{cases} \tag{12}$$

with $\mathcal{X}_n' = \{x_n' : d' \leq x_n' \leq d\}$ and $\mathcal{Y}_n' = \{y_n' : |y_n'| \leq D\}$ and tuning parameter $l_s$ that defines the slope of the (linearly increasing) local mixing length parameter. In fact, this parameter could be related to turbulence intensity, *i.e.*, the amount of wake recovery. In this work we will not investigate this relation further. With the above formulation, the number of tuning variables that belong to the turbulence model $(l_s, d, d')$ is reduced to $3\aleph$. Additionally, we assume that $l_s$, $d$ and $d'$ are equal for each turbine in the farm, which reduces the amount of tuning variables that belong to the turbulence model to 3, a quantity that

could be dealt with by an online estimator.

## 2.2   Turbine model

Turbines are modelled using a classical non–rotating actuator disk model (ADM). In this method, each wind turbine is represented by a uniformly distributed force acting on the grid–points where the rotor disk is located. Figure 4 depicts a schematic top–view representation of a turbine with yaw angle $\gamma$.

Using such an approach, the force exerted by the turbines can be expressed as

$$\boldsymbol{f} = \sum_{n=1}^{\aleph} \boldsymbol{f}_n, \quad \text{with} \quad \boldsymbol{f}_n = \frac{c_f}{2} C_{T_n}' [U_n \cos(\gamma_n)]^2 \begin{pmatrix} \cos(\gamma_n + \varphi) \\ \sin(\gamma_n + \varphi) \end{pmatrix} \text{H} \left[ \frac{D}{2} - ||\boldsymbol{s} - \boldsymbol{t}_n||_2 \right] \delta \left[ (\boldsymbol{s} - \boldsymbol{t}_n) \cdot \boldsymbol{e}_{\perp,n} \right], \tag{13}$$

with H$[\cdot]$ the Heaviside function, $\delta[\cdot]$ the Dirac delta function, $\boldsymbol{e}_{\perp,n}$ the unit vector perpendicular to the $n^{\text{th}}$ rotor disk with position $\boldsymbol{t}_n$. Furthermore more we have $C_{T_n}'$ the disk–based thrust coefficient following (Meyers and Meneveau, 2010), which





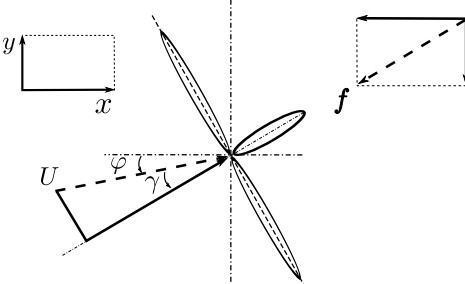

**Figure 4.** Schematic representation of a turbine with yaw angle $\gamma_n$ and flow velocity $U = \left([u(x_n,y_n)]^2 + [v(x_n,y_n)]^2\right)^{1/2}$ at the rotor. Figure adapted from (Jiménez et al., 2010).

can be expressed in terms of the classical thrust coefficient $C_{T_n}$ using the following relation: $C'_{T_n} = C_{T_n}/(1 - a_n)^2$ with $a_n$ the axial induction factor of the $n^{\text{th}}$ turbine. Interestingly, the coefficient $C'_{T_n}$ can directly be related to the turbine set–point in terms of blade pitch angle and rotational speed (see, *e.g.*, Appendix A in (Goit and Meyers, 2015)). In the WFSim model, $C'_{T_n}$ and yaw angle $\gamma_n$ are considered as the control variables and can thus be used to regulate the wakes and hence wind farm

performance. Furthermore, the scalar $c_f$ in (13) can be regarded as a tuning variable and will in this work be set equal for all turbines in the farm.

### 2.3   Power

From the resolved flow velocity components, the power generated by the farm is computed as:

$$P = \sum_{n=1}^{\aleph} \frac{1}{2}\rho A C_{P_n}[U_n\cos(\gamma_n)]^3, \tag{14}$$

It is stated in (Goit and Meyers, 2015) (Appendix A) that when there is no drag and swirl is added to the wake, $C'_{T_n} = C_{P_n}$. Since this is an idealized situation, a loss factor will be introduced such that $C_{P_n} = c_p C'_{T_n}$. The scalar $c_p$ can be seen as a tuning variable and will be set equal for all turbines in the farm. In the above power expression, we have the factor $\cos(\gamma_n)^3$ with exponent 3. In literature such as, *e.g.*, (Gebraad et al., 2014a) and (Medici, 2005) (page 37), numerical values for the exponent were given according to LES and wind tunnel data, respectively. However, to date, the exact value for it is still under

research and since this is outside the scope of this study, the value of the exponent will be three.

This concludes the formulation of the WFSim model. In order to resolve for flow velocity components and wind farm power, the governing equations given in (7) and (8) need to be discretized, a topic that will be discussed in the following section.

### 2.4   Discretization

The set of equations are spatial discretized over a staggered grid following (Versteeg and Malalasekera, 2007). It is carried out

by employing the Finite Volume Method and the Hybrid Differencing scheme. Temporal discretization is performed using the



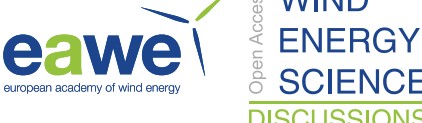

implicit method that is unconditionally stable (Versteeg and Malalasekera, 2007). This boils down to deriving the integrals:

$$\int\limits_{\Delta t}\int\limits_{\Delta V}\left[\frac{\partial \boldsymbol{u}}{\partial t}+(\boldsymbol{u}\cdot\nabla_H)\boldsymbol{u}+\nabla_H\cdot\boldsymbol{\tau}_H+\nabla_Hp-\boldsymbol{f}\right]\mathrm{d}V\,\mathrm{d}t=0,$$

$$\int\limits_{\Delta t}\int\limits_{\Delta V}\left[\nabla_H\cdot\boldsymbol{u}+\frac{\partial v}{\partial y}\right]\mathrm{d}V\,\mathrm{d}t=0,$$

(15)

with $\Delta V$ the volume of one cell (see Fig. 5) and $\Delta t$ the sample period. One obtains, for each cell, the following fully discretized Navier–Stokes equations (for detailed derivation we refer to Appendix A):

– x-momentum equation for the $(i,J)^{\text{th}}$ cell (black in Fig. 5):

$$a_{i,J}^{px}u_{i,J}=\begin{pmatrix}a_{i,J}^{nx} & a_{i,J}^{sx} & a_{i,J}^{wx} & a_{i,J}^{ex}\end{pmatrix}\begin{pmatrix}u_{i,J+1} & u_{i,J-1} & u_{i-1,J} & u_{i+1,J}\end{pmatrix}^T-\delta y_{j,j+1}\left(p_{I,J}-p_{I-1,J}\right)+f_{i,J}^x+\ldots$$

$$\ldots+\begin{pmatrix}a_{i,J}^{nwx} & a_{i,J}^{swx} & a_{i,J}^{nex} & a_{i,J}^{sex}\end{pmatrix}\begin{pmatrix}v_{I-1,j+1} & v_{I-1,j} & v_{I,j+1} & v_{I,J}\end{pmatrix}^T$$

(16)

       – y-momentum equation for the $(I,j)^{\text{th}}$ cell (yellow in Fig. 5):

$$a_{I,j}^{py}v_{I,j}=\begin{pmatrix}a_{I,j}^{ny} & a_{I,j}^{sy} & a_{I,j}^{wy} & a_{I,j}^{ey}\end{pmatrix}\begin{pmatrix}v_{I,j+1} & v_{I,j-1} & v_{I-1,j} & v_{I+1,j}\end{pmatrix}^T-\delta x_{i,i+1}\left(p_{I,J}-p_{I,J-1}\right)+f_{I,j}^y+\ldots$$

$$\ldots+\begin{pmatrix}a_{i,J}^{nwy} & a_{i,J}^{swy} & a_{i,J}^{ney} & a_{i,J}^{sey}\end{pmatrix}\begin{pmatrix}u_{i,J} & u_{i,J-1} & u_{i+1,J} & u_{i+1,J-1}\end{pmatrix}^T$$

(17)

       – continuity equation for the $(I,J)^{\text{th}}$ cell (pink in Fig. 5):

$$0=\delta y_{j,j+1}\left(u_{i+1,J}-u_{i,J}\right)+2\delta x_{i,i+1}\left(v_{I,j+1}-v_{I,j}\right),$$

(18)

The states $u_{\bullet,\bullet},v_{\bullet,\bullet},p_{\bullet,\bullet}$ are defined for the time $k+1$ while the coefficients $a_{\bullet,\bullet}^{\bullet}$ and the forcing terms $f_{\bullet,\bullet}^{\bullet}$ depend on the state at time $k$. Detailed definitions of these coefficients are given in Appendix A, Table 5. Note in (18), the appearance of the previously explained factor 2 (see (8)).



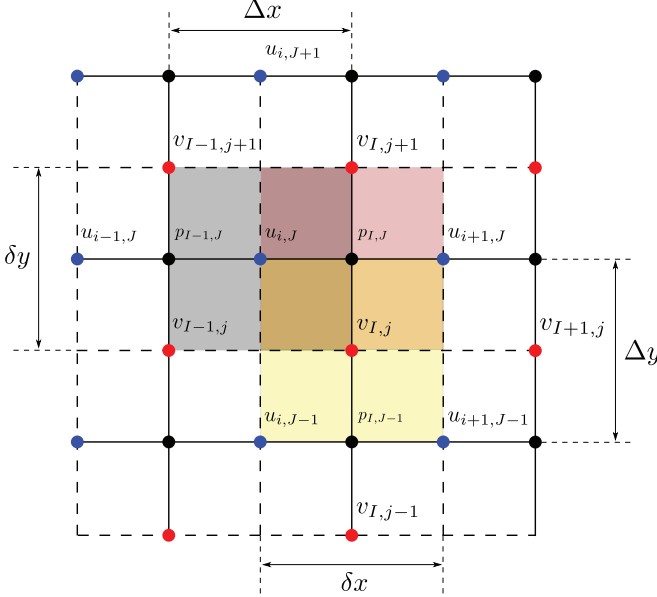

**Figure 5.** One cell for the x–momentum equation (grey with in its centre $u_{i,J}$), one for the y–momentum equation (yellow with in its centre $v_{I,j}$) and one for the continuity equation (pink with in its centre $p_{I,J}$). All three cells have equal dimensions and overlap.

Next, the state vectors $u_k, v_k$ and $p_k$ and control variable vectors $\nu_k$ and $\gamma_k$ at time step $k$ will be defined:

$$
u_k = \begin{pmatrix} u_{3,2} \\ \vdots \\ u_{3,N_y-1} \\ u_{4,2} \\ \vdots \\ u_{4,N_y-1} \\ \vdots \\ u_{N_x-1,2} \\ \vdots \\ u_{N_x-1,N_y-1} \end{pmatrix}, \quad
v_k = \begin{pmatrix} v_{2,3} \\ \vdots \\ v_{2,N_y-1} \\ v_{3,3} \\ \vdots \\ v_{3,N_y-1} \\ \vdots \\ v_{N_x-1,3} \\ \vdots \\ v_{N_x-1,N_y-1} \end{pmatrix}, \quad
p_k = \begin{pmatrix} p_{2,2} \\ \vdots \\ p_{2,N_y-1} \\ p_{3,2} \\ \vdots \\ p_{3,N_y-1} \\ \vdots \\ p_{N_x-1,3} \\ \vdots \\ p_{N_x-1,N_y-2} \end{pmatrix}, \quad
\nu_k = \begin{pmatrix} C'_{T_1} \\ C'_{T_2} \\ \vdots \\ C'_{T_\aleph} \end{pmatrix}, \quad
\gamma_k = \begin{pmatrix} \gamma_1 \\ \gamma_2 \\ \vdots \\ \gamma_\aleph \end{pmatrix}, \tag{19}
$$

with $N_x$ and $N_y$ the number of cells in the x– and y–direction, respectively, and $\aleph$ the number of turbines in the wind farm. Each component in $u_k, v_k$ and $p_k$ represents a flow velocity and pressure, respectively at a point in the field defined by the
5   subscript. For clarity reasons, an example of a staggered grid is depicted in Fig. 6.



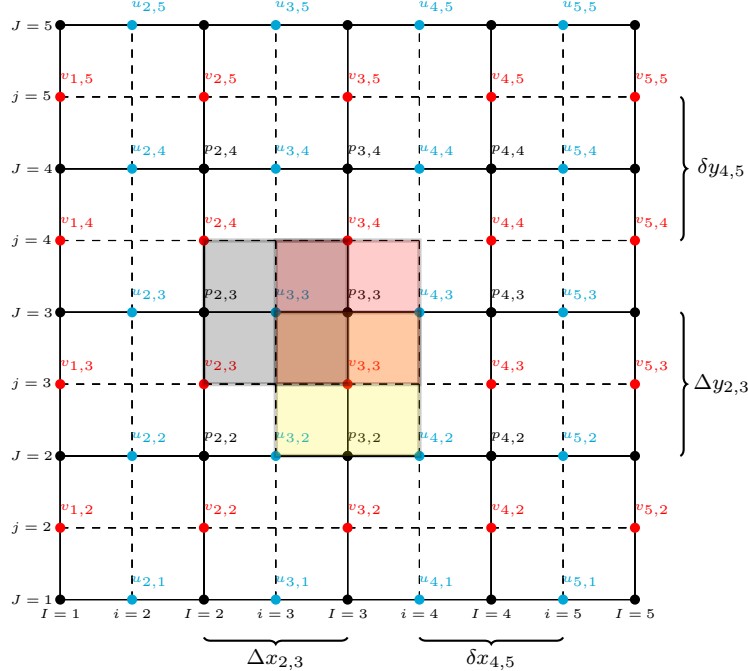

**Figure 6.** Example of a staggered grid with cells each having volume $\Delta V$. In WFSim, the grid is of the type quadrilateral.

## 2.5 Boundary and initial conditions

All the components that are not contained in the vector $u_k, v_k$ and $p_k$, but do appear in the staggered grid need to be defined. For the flow velocity components, first order conditions on the west side of the grid are prescribed assuming the wind is coming from the east. These Dirichlet inflow boundary conditions are related to the ambient inflow defined as $u_b$ and $v_b$ and can vary 5 over time. Zero stress (also referred to as Neumann) boundary conditions are prescribed on the other boundaries. Therefore, for the flow velocity components on the boundaries we define:

$$
\begin{aligned}
u_{2,J} &= u_b & \text{for } J &= 1,2,\ldots,N_y, & v_{1,j} &= v_b & \text{for } j &= 2,3,\ldots,N_y, \\
u_{i,N_y} &= u_{i,N_y-1} & \text{for } i &= 3,4,\ldots,N_x, & v_{I,N_y} &= v_{I,N_y-1} & \text{for } I &= 2,3,\ldots,N_x, \\
u_{i,1} &= u_{i,2} & \text{for } i &= 3,4,\ldots,N_x, & v_{I,2} &= v_{I,3} & \text{for } I &= 2,3,\ldots,N_x, \\
u_{N_x,J} &= u_{N_x-1,J} & \text{for } J &= 2,3,\ldots,N_x-1, & v_{N_x,j} &= v_{N_x-1,j} & \text{for } j &= 3,4,\ldots,N_y-1.
\end{aligned}
$$

For the initial conditions, we define all flow velocity components in the field as $u_b$ and $v_b$, respectively, the boundary velocity components. The initial pressure field is set to zero. Note that by defining the boundary conditions as given above, the assumption is that the wind is coming from the east in Fig. 6, which coincides with the definition of the mixing length (see





§2.1). Finally, the equations given in (7) and (8) can be transformed to the difference algebraic equation (DAE): [1]

$$
\underbrace{\begin{pmatrix} A_x(u_k,v_k) & A_{xy}(u_k) & B_1 \\ A_{yx}(u_k) & A_y(u_k,v_k) & B_2 \\ B_1^T & 2B_2^T & 0 \end{pmatrix}}_{E(q_k)} \underbrace{\begin{pmatrix} u_{k+1} \\ v_{k+1} \\ p_{k+1} \end{pmatrix}}_{q_{k+1}} = \underbrace{\begin{pmatrix} A_{11} & 0 & 0 \\ 0 & A_{22} & 0 \\ 0 & 0 & 0 \end{pmatrix}}_{A} \underbrace{\begin{pmatrix} u_k \\ v_k \\ p_k \end{pmatrix}}_{q_k} + \underbrace{\begin{pmatrix} b_1(u_k,v_k,\nu_k,\gamma_k) \\ b_2(u_k,v_k,\nu_k,\gamma_k) \\ b_3 \end{pmatrix}}_{b(q_k,w_k)},
\tag{20}
$$

with $n_q = n_u + n_v + n_p$ and $u_k \in \mathbb{R}^{n_u}, v_k \in \mathbb{R}^{n_v}, p_k \in \mathbb{R}^{n_p}$ containing all flow velocities in the longitudinal and lateral direction and the pressure vector at time $k$, respectively, and control variable $w_k = \begin{pmatrix} \nu_k^T & \gamma_k^T \end{pmatrix}^T \in \mathbb{R}^{2\aleph}$. The non–singular square descriptor matrix $E(q_k)$ contains the coefficients $a_{\bullet,\bullet}^{\bullet}$, appearing in (16) and (17), that depend on the state at time $k$. The square constant matrix $A$ solely depends on grid spacing and sample period $\Delta t$. Note that the state vector contains three states for every cell hence an increase in grid resolution results in an increase in matrix dimensions. However, the system matrices that occur in (20) are sparse and efficient numerical solvers are available for these kind of problems. This will be demonstrated in § 2.6. The vector $b(q_k,w_k)$ contains the forcing terms (turbines) and boundary conditions.

By defining $N_x, N_y, \Delta x_{I,I+1}, \Delta y_{J,J+1}$ and the turbine positions, a wind farm topology is determined. Next, ambient flow conditions $u_b$ and $v_b$, tuning parameters $c_f, c_p, d, d', l_s$ and the control variable $w_k$ need to be specified. The system given in (20) is then fully defined and can be solved.

## 2.6 Computation time

When discretizing partial differential equations, a trade–off has to be made between the computation time and grid resolution. Typically, a higher resolution results in more precise computation of the variables, but also increasing computation time. In WFSim, computational cost is reduced by exploiting sparsity and by applying the Reverse Cuthill–McKee algorithm (George and Liu, 1981).[2] The latter is applicable due to the fact that the matrix structure is fixed. The interested reader is referred to (Doekemeijer et al., 2016) for more information on the Cuthill–McKee algorithm in WFSim.

In this section, the mean computation time needed for one time step $\Delta t^{\text{cpu}}$ will be analysed. The presented results are obtained on a regular notebook employing one core. Since the objective is to do online control, *i.e.*, it is desired to reduce computational complexity, this section introduces a second WFSim representation. The first representation was given in (20) while the second is defined as:

$$
\underbrace{\begin{pmatrix} A_x(u_k,v_k) & 0 & B_1 \\ 0 & A_y(u_k,v_k) & B_2 \\ B_1^T & 2B_2^T & 0 \end{pmatrix}}_{E(q_k)} \underbrace{\begin{pmatrix} u_{k+1} \\ v_{k+1} \\ p_{k+1} \end{pmatrix}}_{q_{k+1}} = \underbrace{\begin{pmatrix} A_{11} & 0 & 0 \\ 0 & A_{22} & 0 \\ 0 & 0 & 0 \end{pmatrix}}_{A} \underbrace{\begin{pmatrix} u_k \\ v_k \\ p_k \end{pmatrix}}_{q_k} + \underbrace{\begin{pmatrix} b_1(u_k,v_k,\nu_k,\gamma_k) \\ b_2(u_k,v_k,\nu_k,\gamma_k) \\ b_3 \end{pmatrix}}_{b(q_k,w_k)}.
\tag{21}
$$

The difference can be found in the descriptor matrix. In the above representation, the elements $A_{xy}(u_k), A_{yx}(u_k)$ that occur in (20) are set to zero. This can be justified by the fact that these matrices contain elements that, for our case studies, are relatively small hence their contribution is negligible. Therefore, no significant change in the flow field computation has been

---

[1]This type of system can also be referred to as a quasi linear parameter varying model or descriptor model.
[2]The sparse toolbox and reverse Cuthill–McKee algorithm are both utilised in Matlab.

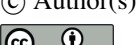



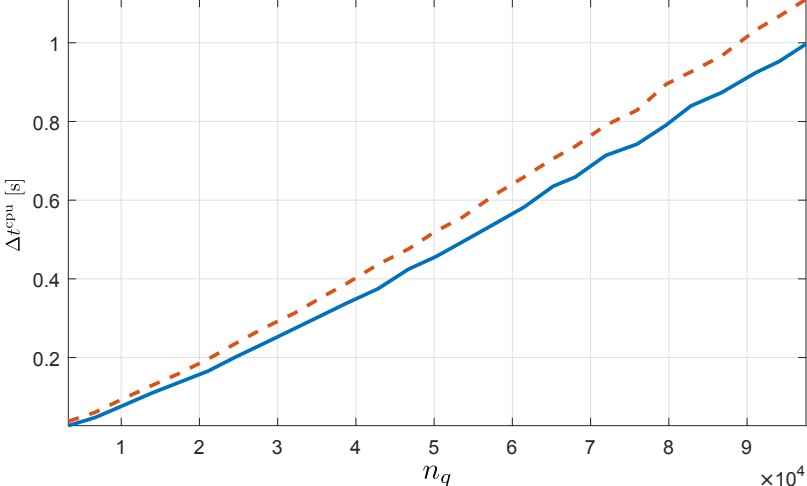

**Figure 7.** Mean computation time per simulation time step $\Delta t^{\text{cpu}}$ versus number of states $n_q$. Red dashed is WFSim as presented in (20) and blue is WFSim as presented in (21). Note that the number of cells is approximately $n_q/3$ with $n_q$ the number of states.

observed, but a decrease in $\Delta t^{\text{cpu}}$ (see Fig. 7), the remainder of this paper will continue with the WFSim representation given in (21). Table 2 depicts more numerical values of $\Delta t^{\text{cpu}}$ for this WFSim representation.

**Table 2.** Mean computation time per simulation time step $\Delta t^{\text{cpu}}$ versus number of states $n_q$ for the WFSim representation as given in (21). Computation are done on a regular note book on one core.

| $n_q$ | $\Delta t^{\text{cpu}}$ [s] | $n_q$ | $\Delta t^{\text{cpu}}$ [s] | $n_q$ | $\Delta t^{\text{cpu}}$ [s] | $n_q$ | $\Delta t^{\text{cpu}}$ [s] |
|---|---|---|---|---|---|---|---|
| $3 \cdot 10^3$ | 0.02 | $27 \cdot 10^3$ | 0.22 | $115 \cdot 10^3$ | 1.19 | $239 \cdot 10^3$ | 3.1 |
| $6 \cdot 10^3$ | 0.04 | $43 \cdot 10^3$ | 0.37 | $147 \cdot 10^3$ | 1.66 | $258 \cdot 10^3$ | 3.5 |
| $9 \cdot 10^3$ | 0.06 | $64 \cdot 10^3$ | 0.60 | $182 \cdot 10^3$ | 2.12 | $268 \cdot 10^3$ | 3.7 |
| $14 \cdot 10^3$ | 0.10 | $88 \cdot 10^3$ | 0.88 | $221 \cdot 10^3$ | 2.50 | $276 \cdot 10^3$ | 3.8 |

From Table 2 we can conclude that $\Delta t^{\text{cpu}}$ increases between quadratic and linear with respect to the number of states $n_q$ for $n_q < 221 \cdot 10^3$. It depends on the computer properties how much you can increase the number of states until the CPU is out of memory.

## 3  Simulation results

In this section, WFSim flow and power data will be compared against LES data and it is organised as follows. In §3.1, quality measures are introduced. In §3.2.1, WFSim data is compared with PALM data and in §3.2.2, WFSim is validated against SOWFA data. In both simulation cases, the thrust coefficients $C'_T$ is varied while the yaw angles are set to zero.



## 3.1 Quality measures

Suppose we have at time $k$ a measurement of one quantity $z_k \in \mathbb{R}^N$ and its estimation $\hat{z}_k \in \mathbb{R}^N$. Define the prediction error $e_k = \hat{z}_k - z_k$. The quality measure Root Mean Squared Error (RMSE) is, for time step $k$, defined as:

$$\mathrm{RMSE}(z_k, \hat{z}_k) = \sqrt{\frac{1}{N} e_k^T e_k}, \tag{22}$$

This measure is used to compare the flow centreline velocity $U_k^c(x)$ and power signals from LES and WFSim data for different model parameters. The flow centreline is, for one time step, defined as the laterally–averaged longitudinal flow velocity throughout the simulation domain across the rotor diameter. Mathematically this can, for LES data at time step $k$ at longitudinal position $x_i$, be defined as:

$$U_k^c(x_i) = \frac{1}{N_{\overline{y}}} \sum_{s=1}^{N_{\overline{y}}} u_k(x_i, y_s), \tag{23}$$

with $y_s$ the y–coordinate of one cell across the line $\overline{y} \subset y$, which contains $N_{\overline{y}}$ number of cells and having an equal length as the rotor diameter. From WFSim data, the flow velocity component at the rotor centre will be taken accros the position $x$.

In this work we compare lateral and longitudinal flow velocity components at hub height and power signals calculated with LES with lateral and longitudinal flow velocity components and power signals calculated with WFSim.[3]

## 3.2 Axial induction actuation

Studies such as (Shapiro et al., 2017a), (Munters and Meyers, 2017), (Vali et al., 2017) and (van Wingerden et al., 2017) illustrate that axial induction actuation can be used in active power control where the objective is to provide grid facilities. In order to utilize the WFSim model in active power control, it is important to first validate it when exciting the thrust coefficient.

In the following, WFSim is compared against simulation data from PALM (Maronga et al., 2015) and SOWFA (Churchfield et al., 2012), both high–fidelity wind farm models that were briefly discussed in Section 1. The latter includes the actuator line model (ALM) while the former employs the ADM.[4]

### 3.2.1 PArallelized LES Model (PALM) and WFSim

PALM predicts the 3D flow velocity vectors and turbine power signals in a wind farm using LES and is based on the 3D incompressible Navier–Stokes equations.[5] Table 3 gives a summary of the 2–turbine wind farm simulated in WFSim. A summary of the PALM simulation set–up can be found in Appendix B. The applied control signals are depicted in Fig. 8 and are chosen such that different system dynamics are excited. The final values for the tuning parameters are obtained using a grid search. Figure 9 and Fig. 10 show a comparison of the mean flow centreline and the wind farm power, respectively. A flow field evaluated with both the WFSim model and PALM can be found in Appendix B.

---

[3]The LES flow data is mapped onto the grid of WFSim using bilinear interpolation techniques.

[4]PALM also includes the rotating ADM, but in our case study, the ADM is employed.

[5]In this work we consider PALM as a wind farm model since PALM is simulated with turbine models. However, PALM is also applicable for simulating oceanic behaviour.



**Table 3.** Summary of the WFSim simulation set–up.

| | | | |
|---|---|---|---|
| Domain size $L_x \times L_y$ | $2 \times 0.63$ [km$^2$] | Turbine rotor diameter $D$ | 126.4 [m] |
| Grid size $N_x \times N_y$ | $50 \times 25$ | Turbine arrangement | $2 \times 1$ |
| Cell size $\Delta x \times \Delta y$ | $40 \times 23$ [m$^2$] | Turbine spacing | $5D$ |
| Sample period $\Delta t$ | 1 [s] | Atmospheric conditions | $u_b = 8, v_b = 0$ [m/s], $\rho = 1.2$ [kg/m$^3$] |
| Force and power factor | $c_f = 1.7, c_p = 0.95$ | Turbulence model | $d = 530, d' = 122$ [m] $l_s = 0.06$ |

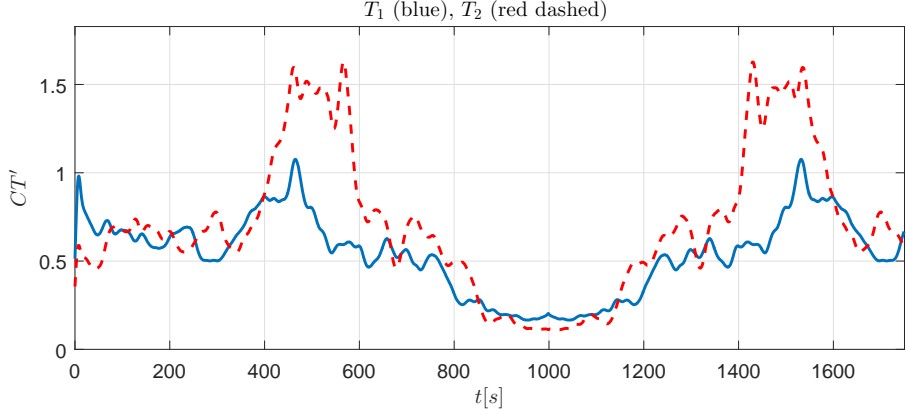

**Figure 8.** Excitation signals for the 2–turbine simulation case. The yaw angles are set to zero.

In Fig. 9, the mean flow centreline through the farm of WFSim and PALM are relatively similar. The PALM data exhibits more turbulent fluctuations due to the presence of a more sophisticated turbulence model, which allows for better capturing small-scale dynamics such as turbine induced turbulence. However, the WFSim model is capable of estimating similar wake recovery as the PALM model. The recovery in the WFSim model is due to the turbulence model as presented in §2.1. It is in fact

the slope of the local mixing length parameters that can determine the amount of wake recovery or more precise, the larger this slope, the more wake recovery will be observed. It is therefore interesting to link this tuning variable to the turbulence intensity in the farm. Furthermore, it can be seen in Fig. 10 that the WFSim model is capable of estimating the wind farm power. Since both the WFSim model and PALM employ the ADM, fast fluctuations in the power signal can be observed. This is due to the lack of rotor inertia in both simulation cases. The simulation case presented in this section illustrates that the WFSim model,

in which the third dimension is partially neglected, is able to estimate wind farm flow and power signals computed with a 3D LES wind farm model. In §3.2.2, a SOWFA case study will be presented, a LES model that includes turbine dynamics.

### 3.2.2 Simulator fOr Wind Farm Applications (SOWFA) and WFSim

SOWFA predicts the 3D flow velocity vectors in a wind farm using LES and is based on the 3D incompressible Navier–Stokes equations. For turbine modeling it employs the actuator line model (ALM), which is a more sophisticated model than

the ADM (Sanderse et al., 2011). In addition, the Fatigue, Aerodynamics, Structures and Turbulence (FAST) model from

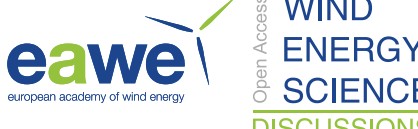



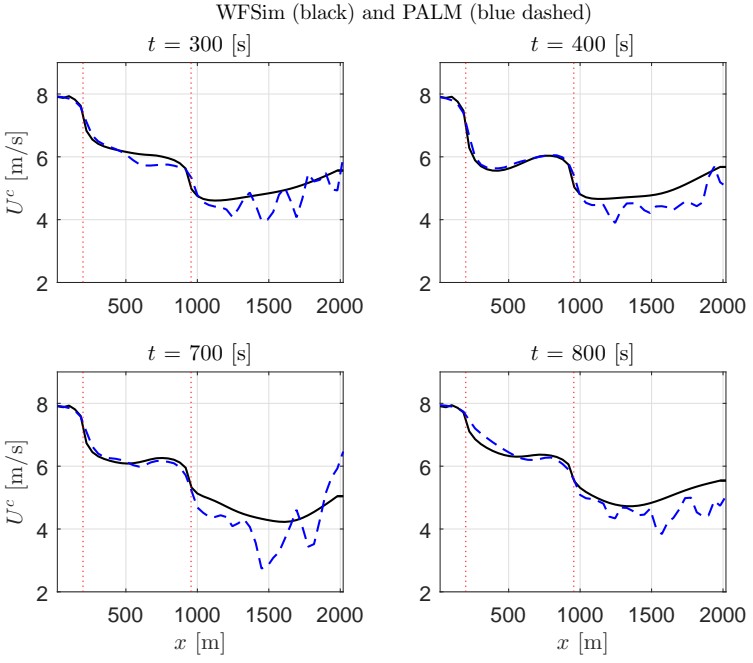

**Figure 9.** Mean flow centreline at four time instances through the farm. The vertical red dashed lines indicate the positions of the turbines.

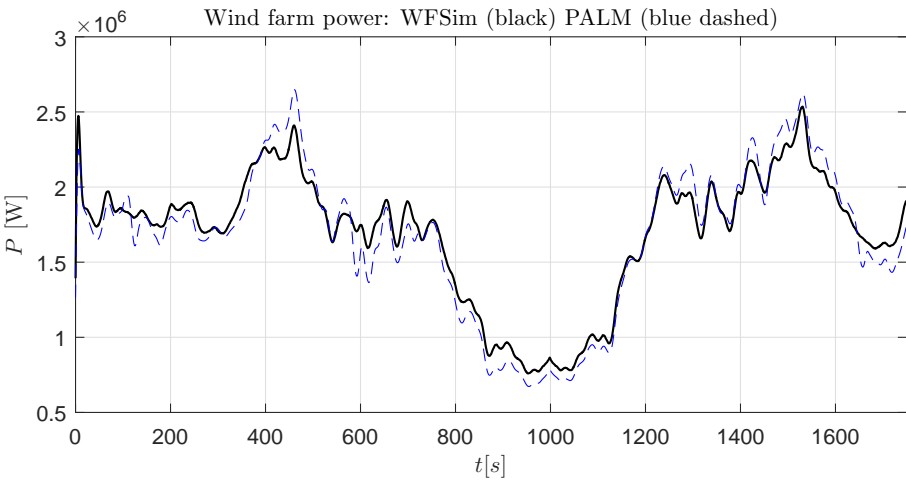

**Figure 10.** Wind farm power from PALM (blue dashed) and WFSim (black).

NREL is implemented (Jonkman and Buhl, 2005). This model calculates, *i.a.*, turbine power production, blade forces on the flow and structural loading on the turbine. In the presented SOWFA simulation, the NREL 5–MW wind turbine is simulated (Jonkman et al., 2009).





The SOWFA data set used in this work for validation is equivalent to the set used in (van Wingerden et al., 2017). The thrust coefficient $C_T'$ is not a control variable in SOWFA due to the employment of the ALM and therefore has to be estimated. This will be discussed in the following paragraph.

**Turbine operating settings**

For estimating the control signals $C_{T_n}'$, the turbine's fore–aft bending moment $M_k^{\text{sowfa}}$ calculated by FAST is exploited. Using the relation $M_k^{\text{sowfa}} = F_k^{\text{sowfa}} z_h$ with $z_h$ the hub height, the (indirect) measured thrust force $F_k^{\text{sowfa}}$ can be derived. An estimation from SOWFA data of the rotor flow velocity $U_k^{\text{sowfa}}$ is obtained by averaging the flow velocity components across the rotor. Using the standard ADM yields for each turbine:

$$F_k^{\text{sowfa}} = \frac{1}{2} A \rho C_T' \left[ U_k^{\text{sowfa}} \right]^2 \begin{pmatrix} \cos(\gamma_k + \varphi_k) \\ \sin(\gamma_k + \varphi_k) \end{pmatrix}. \tag{24}$$

Since $F_k^{\text{sowfa}}, U_k^{\text{sowfa}}$ and $\rho$ can be obtained from SOWFA data and the yaw angles are given, all the variables in (24) are known hence the control variable $C_T'$ can for each turbine be estimated from SOWFA data.[6] It will be used, together with the yaw angle, as an input to the WFSim model.

In the following, flow data at hub height from a 9–turbine SOWFA simulation case will be compared with WFSim data. See Fig. 12 (a) for the simulated wind farm topology. The turbines are excited with thrust coefficients as depicted in Fig. 11. These
excitation signals are estimated from SOWFA data using the relation defined in (24). Table 4 presents the WFSim parameters used during simulations. The tuning variables of the WFSim model are found using a grid–search and the inflow conditions $u_b, v_b$ are estimated from SOWFA data.

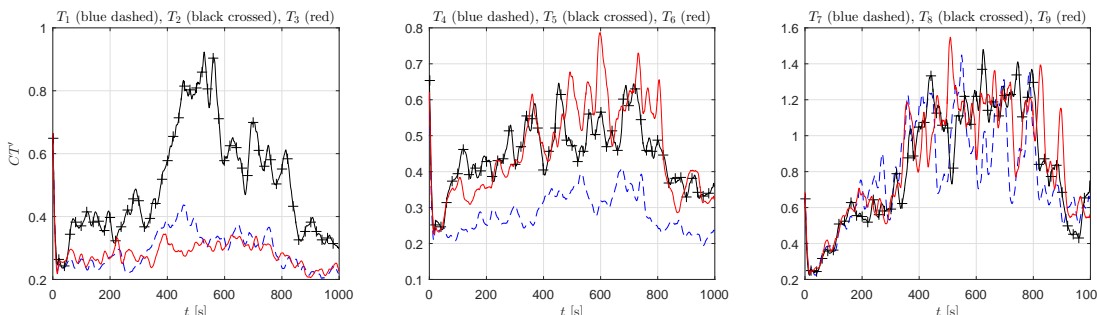

**Figure 11.** Excitation signals for the 9–turbine simulation case. The yaw angels are set to zero.

---

[6]The estimated $C_T'$ from SOWFA data is relatively noisy hence filtered.



**Table 4.** Summary of the WFSim simulation set–up.

| | | | |
|---|---|---|---|
| Domain size $L_x \times L_y$ | $2.5 \times 1.5$ [km$^2$] | Turbine rotor diameter $D$ | $126.4$ [m] |
| Grid size $N_x \times N_y$ | $100 \times 42$ | Turbine arrangement | $3 \times 3$ |
| Cell size $\Delta x \times \Delta y$ | $25 \times 15$ [m$^2$] | Turbine spacing | $5D \times 3D$ |
| Sample period $\Delta t$ | $1$ [s] | Atmospheric conditions | $u_b = 12, v_b = 0$ [m/s], $\rho = 1.2$ [kg/m$^3$] |
| Force and power factor | $c_f = \frac{5}{2}, c_p = 1.1$ | Turbulence model | $d = 635, d' = 76.2$ [m] $l_s = 0.17$ |

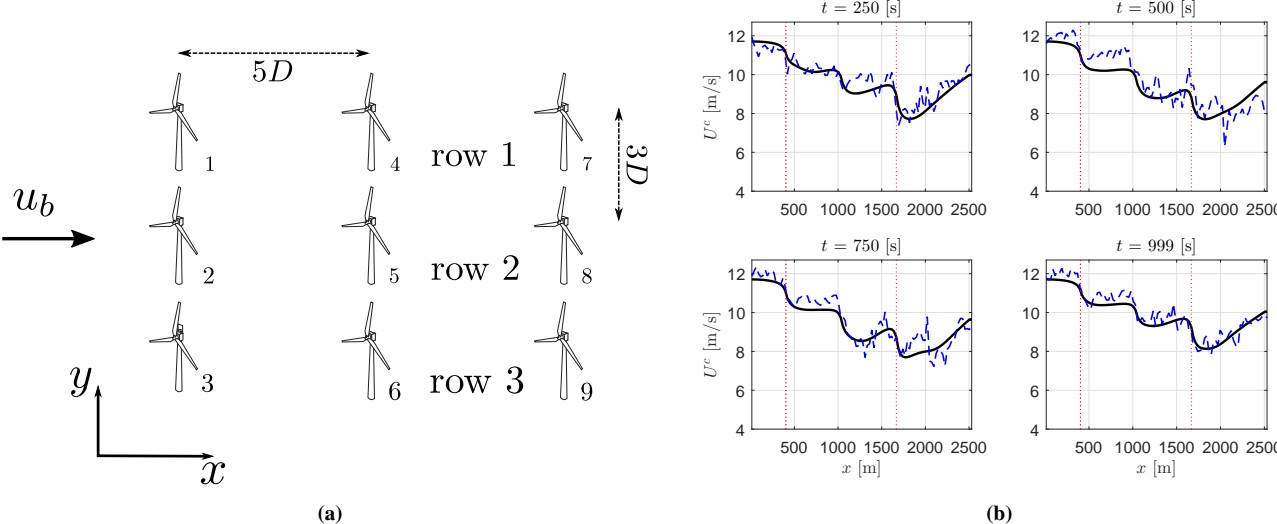

**Figure 12.** Topology simulated wind farm (a) and mean flow centreline at four time instances through the first row (b).

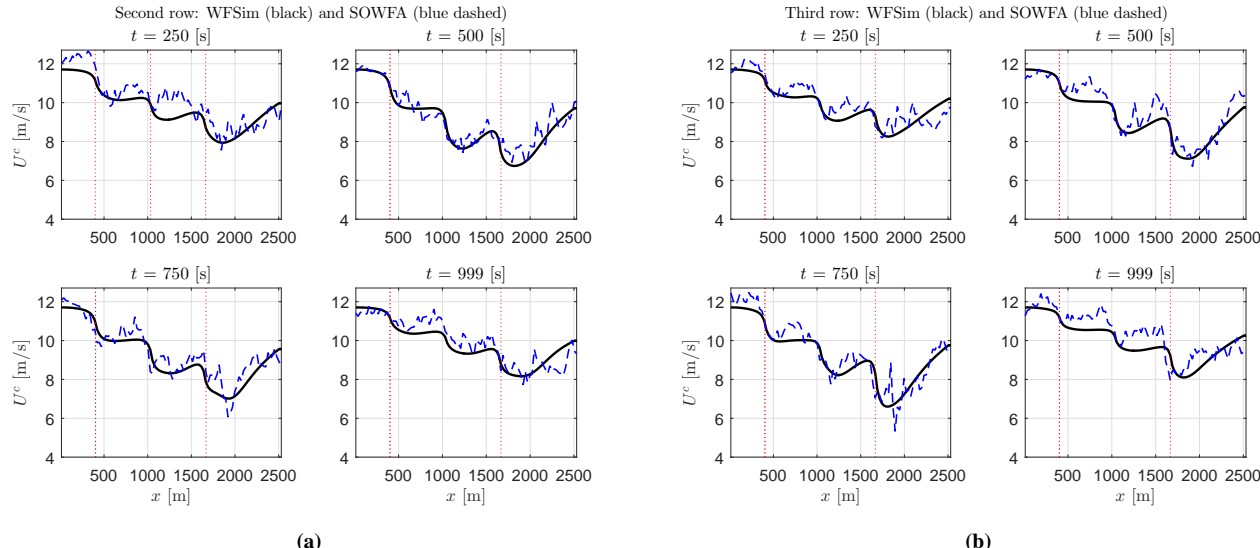

**Figure 13.** Mean flow centreline at four time instances through the second row (a) and third row (b) of turbines. The vertical red dashed lines indicate the positions of the turbines.





Figure 12 (b) and Fig. 13 depict a mean flow centreline (see (23)) comparison for each row at four time instances. It can be concluded that the mean flow centreline derived from WFSim data approximates the mean flow centreline derived from SOWFA data. In Fig. 14, time series of the power signals from SOWFA and WFSim are depicted. The signals from the latter are more oscillating than the power signals from SOWFA. This is due to the fact that the power expression in WFSim is a nonlinear static map depending on the $C'_T$. Thus, no turbine dynamics are taken into account, which is contrary to SOWFA in which the FAST turbine model is simulated. However, important characteristics can be captured with WFSim. A flow field evaluated with both the WFSim model and SOWFA can be found in Appendix C.

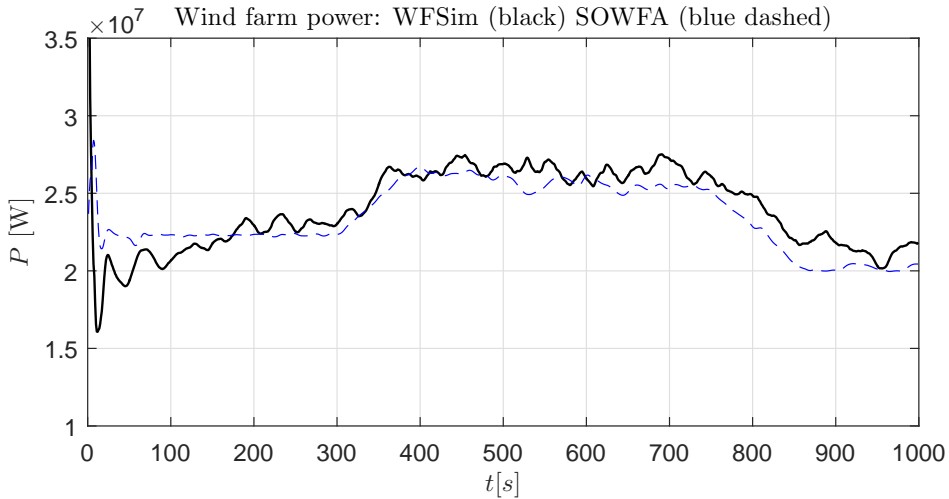

**Figure 14.** Wind farm power from SOWFA (blue dashed) and WFSim (black).

WFSim is capable of estimating dominant wake dynamics, the objective of the control–oriented model WFSim. Smaller scale and stochastic effects can be measured by sensors and incorporated using an estimator based on WFSim, as has been shown in (Doekemeijer et al., 2016, 2017).



## 4    Conclusions

Current literature on wind farm control can be categorized in model free and model based methods. This manuscript focused on the latter category. In here, a distinction can be made between employed type of model, a steady–state or dynamic wind farm model. In order to use the closed–loop control paradigm, and account for model uncertainties, we think it is important to employ

a dynamic wind farm model for controller design and possible online wind farm control. In this paper, such a control–oriented dynamic wind farm model, referred to as WFSim, has been presented.[7] It is a wind farm model that can predict flow fields and power production and includes turbines that are modelled using actuator disk theory and is based on modified two–dimensional Navier–Stokes equations. Completely neglecting the third (vertical) dimension is a too crude assumption to describe accurately enough the flow in a wind farm for control purposes. In this paper, we included a correction term in the continuity equation.

It has been illustrated that the inclusion of this factor reduces the effect of neglecting the third (vertical) dimension. More precisely, it has been shown that the speed–up effect of the flow on the right and left downwind of a turbine will be reduced when solving for the corrected Navier–Stokes equations compared to the standard two–dimensional Navier–Stokes equations. It has been shown that this resulted in a better approximation of LES data.

In addition, a turbulence model was included taking into account the desired wake recovery. The turbulence model is based

on Prandtl's mixing length hypotheses, where the mixing length parameter is made dependent on the downstream distance from the turbine rotors and also dependent on the mean wind direction. After theoretically formulating the WFSim model, this paper followed by illustrating that the computed flow velocities and power signals from the 2D–like WFSim model can estimate flow velocity data and power signals from the 3D high–fidelity wind farm models PALM and SOWFA. The necessary computation time of the WFSim model is however a fraction of what is needed to do LES making the WFSim model suitable for online

control. This work focussed on axial induction actuation, but future work will also include the validation of yaw actuation and wind direction changes. In addition, future work will entail the online update of the tuning variables $c_f, c_p, d, d', l_s$ by an observer and the employment of the presented dynamic wind farm model in an online closed–loop control scheme.

---

[7]The WFSim repository can be found in (https://github.com/TUDelft DataDrivenControl/WFSim).





**Appendix A:  Discretizing the Navier-Stokes equations.**

This section will present the necessary derivations to go from Eq. (15) to Eq. (20), *i.e.*, it will elaborate on the discretization of the NS equations. In the following subsections, all terms in the NS equations will be dealt with subsequently.

**A1    Discretizing the convection (nonlinear) terms**

The nonlinear term that occurs in the momentum equations can be spatially discretized by deriving:

$$
\int_{\Delta V} \rho(\mathbf{u} \cdot \nabla)\mathbf{u} \, dV = \int_{\Delta V} \rho \begin{pmatrix} \frac{\partial u^2}{\partial x} + \frac{\partial uv}{\partial y} \\ \frac{\partial vu}{\partial x} + \frac{\partial v^2}{\partial y} \end{pmatrix} dV.
$$

**x-momentum equation**

Deriving the term in the x-momentum equation (first element in the above vector) yields:

$$
\int_{\Delta V} \rho \left[ \frac{\partial u^2}{\partial x} + \frac{\partial uv}{\partial y} \right] dV = \rho \left[ \left( u^2 \delta y \right)_e - \left( u^2 \delta y \right)_w + (uv\Delta x)_n - (uv\Delta x)_s \right],
$$

where $\left( u^2 \delta y \right)_e, \left( u^2 \delta y \right)_w$ are the quantities $u^2$ at the east and west side of the cell having surface $\delta y_e, \delta y_w$, respectively. Similarly, $(uv\Delta x)_n, (uv\Delta x)_s$ are the quantities $uv$ at the north and south side of the cell having surface $\Delta x_n, \Delta x_s$, respectively. Assuming $\delta y = \delta y_e = \delta y_w$ and $\Delta x = \Delta x_n = \Delta x_s$, the above can be written as:

$$
\int_{\Delta V} \rho \left[ \frac{\partial u^2}{\partial x} + \frac{\partial uv}{\partial y} \right] dV = \rho \left[ \left( u^2 \right)_e \delta y - \left( u^2 \right)_w \delta y + (uv)_n \Delta x - (uv)_s \Delta x \right],
$$

Define $F^{ex} = \rho u_e \delta y, F^{wx} = \rho u_w \delta y, F^{nx} = \rho v_n \Delta x, F^{sx} = \rho v_s \Delta x$. This is in (Versteeg and Malalasekera, 2007) referred to

as a convective mass flux approximation. The above can then be written as:

$$
\int_{\Delta V} \rho \left[ \frac{\partial u^2}{\partial x} + \frac{\partial uv}{\partial y} \right] dV = F^{ex} u_e - F^{wx} u_w + F^{nx} u_n - F^{sx} u_s,
$$

In Fig. 5 we observe that $u_e, u_w, u_n, u_s, v_n, v_s$ are not defined for the black cell. Applying central differencing approximates the terms as follows:

$$
u_e = \frac{u_{i+1,J} + u_{i,J}}{2}, \quad u_w = \frac{u_{i-1,J} + u_{i,J}}{2}, \quad u_n = \frac{u_{i,J+1} + u_{i,J}}{2}, \quad u_s = \frac{u_{i,J-1} + u_{i,J}}{2},
$$

$$
v_n = \frac{v_{I-1,j+1} + v_{I,j+1}}{2}, \quad v_s = \frac{v_{I-1,j} + v_{I,j}}{2}. \tag{A1}
$$

We can now write:

$$
\int_{\Delta V} \rho \left[ \frac{\partial u^2}{\partial x} + \frac{\partial uv}{\partial y} \right] dV = F_{i,J}^{ex} u_{i+1,J} - F_{i,J}^{wx} u_{i-1,J} + F_{i,J}^{nx} u_{i,J+1} - F_{i,J}^{sx} u_{i,J-1} + \left( F_{i,J}^{ex} - F_{i,J}^{wx} + F_{i,J}^{nx} - F_{i,J}^{sx} \right) u_{i,J}.
$$

In Eq. (A1), central differencing is applied. A disadvantage of this method is that it does not use prior knowledge on the flow direction. The upwind differencing scheme however employs this prior knowledge as explained in (Versteeg and Malalasekera,



2007). A combination of the central and upwind differencing scheme is the hybrid differencing scheme. When applying this, the above can be written as:

$$\int\limits_{\Delta V} \rho \left[ \frac{\partial u^2}{\partial x} + \frac{\partial uv}{\partial y} \right] \mathrm{d}V = c_{i,J}^{ex} u_{i+1,J} - c_{i,J}^{wx} u_{i-1,J} + c_{i,J}^{nx} u_{i,J+1} - c_{i,J}^{sx} u_{i,J-1} + c_{i,J}^{px} u_{i,J}, \tag{A2}$$

with $c_{i,J}^{ex} = \max\left[-F_{i,J}^{ex}, 0\right], c_{i,J}^{wx} = \max\left[F_{i,J}^{wx}, 0\right], c_{i,J}^{nx} = \max\left[-F_{i,J}^{nx}, 0\right], c_{i,J}^{sx} = \max\left[F_{i,J}^{sx}, 0\right]$ and $c_{i,J}^{px} = c_{i,J}^{ex} + c_{i,J}^{wx} + c_{i,J}^{nx} +$
$c_{i,J}^{sx} + F_{i,J}^{ex} - F_{i,J}^{wx} + F_{i,J}^{nx} - F_{i,J}^{sx}$. In WFSim, the coefficients $c_{i,J}^{\bullet}$ and $F_{i,J}^{\bullet}$ are evaluated for time $k$ while the other flow velocity components are computed for time $k+1$.

**y-momentum equation**

Deriving the nonlinear term in the y-momentum equation yields:

$$\int\limits_{\Delta V} \rho \left[ \frac{\partial v^2}{\partial y} + \frac{\partial vu}{\partial x} \right] \mathrm{d}V = F_{I,j}^{ey} v_{I+1,j} - F_{I,j}^{wy} v_{I-1,j} + F_{I,j}^{ny} v_{I,j+1} - F_{I,j}^{sy} v_{I,j-1} + \left( F_{I,j}^{ey} - F_{I,j}^{wy} + F_{I,j}^{ny} - F_{I,j}^{sy} \right) v_{I,j},$$

with $F_{I,j}^{ey} = \rho u_e \Delta y, F_{I,j}^{wy} = \rho u_w \Delta y, F_{I,j}^{ny} = \rho v_n \delta x, F_{I,j}^{sy} = \rho v_s \delta x$ and:

$$v_e = \frac{v_{I+1,j} + v_{I,j}}{2}, \quad v_w = \frac{v_{I-1,j} + v_{I,j}}{2}, \quad v_n = \frac{v_{I,j+1} + v_{I,j}}{2}, \quad v_s = \frac{v_{I,j-1} + v_{I,j}}{2},$$
$$u_e = \frac{u_{i+1,J} + u_{i+1,J-1}}{2}, \quad u_w = \frac{u_{i,J} + u_{i,J-1}}{2}.$$

The intermediate steps are omitted here since they are similar to the steps presented when handling the nonlinear term in the x-momentum equation. Note however that the discretization is evaluated using the yellow cell (see Fig. 5). When applying the
hybrid differencing scheme, the above can be written as:

$$\int\limits_{\Delta V} \rho \left[ \frac{\partial v^2}{\partial y} + \frac{\partial vu}{\partial x} \right] \mathrm{d}V = c_{I,j}^{ey} v_{I+1,j} - c_{I,j}^{wy} v_{I-1,j} + c_{I,j}^{ny} v_{I,j+1} - c_{I,j}^{sy} v_{I,j-1} + c_{I,j}^{py} v_{I,j}, \tag{A3}$$

with $c_{I,j}^{ey} = \max\left[-F_{I,j}^{ey}, 0\right], c_{I,j}^{wy} = \max\left[F_{I,j}^{wy}, 0\right], c_{I,j}^{ny} = \max\left[-F_{I,j}^{ny}, 0\right], c_{I,j}^{sy} = \max\left[F_{I,j}^{sy}, 0\right]$ and $c_{I,j}^{py} = c_{I,j}^{ey} + c_{I,j}^{wy} + c_{I,j}^{ny} +$
$c_{I,j}^{sy} + F_{I,j}^{ey} - F_{I,j}^{wy} + F_{I,j}^{ny} - F_{I,j}^{sy}$. Similar as before, the coefficients $c_{I,j}^{\bullet}$ and $F_{I,j}^{\bullet}$ are evaluated for time $k$ while the other flow velocity components are computed for time $k+1$.

**A2    Discretizing the pressure gradient**

For the pressure gradient we evaluate:

$$\int\limits_{\Delta V} \begin{pmatrix} \frac{\partial p}{\partial x} \\ \frac{\partial p}{\partial y} \end{pmatrix} \mathrm{d}V = \begin{pmatrix} (p_{I,J} - p_{I-1,J})\, \delta y \\ (p_{I,J} - p_{I,J-1})\, \delta x \end{pmatrix}.$$

The pressure components are evaluated for time $k+1$.



## A3  Discretizing the stress term

Evaluate:

$$
\int_{\Delta V} \overline{\tau} \nabla \, \mathrm{d}V = \int_{\Delta V} \left( \begin{array}{c} \frac{\partial}{\partial x}\left[ l_u(x,y)^2 \left| \frac{\partial u}{\partial y} \right| \frac{\partial u}{\partial x} \right] + \frac{\partial}{\partial y} \frac{1}{2} \left[ l_u(x,y)^2 \left| \frac{\partial u}{\partial y} \right| \left( \frac{\partial u}{\partial y} + \frac{\partial v}{\partial x} \right) \right] \\ \frac{\partial}{\partial y}\left[ l_u(x,y)^2 \left| \frac{\partial u}{\partial y} \right| \frac{\partial v}{\partial y} \right] + \frac{\partial}{\partial x} \frac{1}{2} \left[ l_u(x,y)^2 \left| \frac{\partial u}{\partial y} \right| \left( \frac{\partial u}{\partial y} + \frac{\partial v}{\partial x} \right) \right] \end{array} \right) \, \mathrm{d}V. \tag{A4}
$$

### x-momentum equation

5   Considering the x-momentum equation we have to evaluate multiple terms. The first term evaluates as:

$$
\int_{\Delta V} \frac{\partial}{\partial x}\left[ l_u(x,y)^2 \left| \frac{\partial u}{\partial y} \right| \frac{\partial u}{\partial x} \right] \, \mathrm{d}V = \left[ l_u(x,y)^2 \left| \frac{\partial u}{\partial y} \right| \frac{\partial u}{\partial x} \right]_e \delta y - \left[ l_u(x,y)^2 \left| \frac{\partial u}{\partial y} \right| \frac{\partial u}{\partial x} \right]_w \delta y.
$$

Here we have:

$$
\left.\frac{\partial u}{\partial y}\right|_e = \frac{u_{i,J+1} - u_{i,J}}{\Delta y_{J,J+1}}, \qquad \left.\frac{\partial u}{\partial x}\right|_e = \frac{u_{i+1,J} - u_{i,J}}{\delta x_{i,i+1}}, \qquad \left.\frac{\partial u}{\partial y}\right|_w = \frac{u_{i,J} - u_{i,J-1}}{\Delta y_{J-1,J}}, \qquad \left.\frac{\partial u}{\partial x}\right|_w = \frac{u_{i,J} - u_{i-1,J}}{\delta x_{i-1,i}},
$$

and $\delta y = \delta y_{j,j+1}$. Substituting these expressions yields:

$$
\int_{\Delta V} \frac{\partial}{\partial x}\left[ l_u(x,y)^2 \left| \frac{\partial u}{\partial y} \right| \frac{\partial u}{\partial x} \right] \, \mathrm{d}V = \underbrace{l_u(x_{I-1},y_J)^2 \left| \frac{(u_{i,J+1} - u_{i,J})\delta y_{j,j+1}}{\Delta y_{J,J+1}\delta x_{i,i+1}} \right|}_{T_{i,J}^{ex}} (u_{i+1,J} - u_{i,J}) \ldots
$$
$$
\underbrace{- l_u(x_I,y_J)^2 \left| \frac{(u_{i,J} - u_{i,J-1})\delta y_{j,j+1}}{\Delta y_{J-1,J}\delta x_{i-1,i}} \right|}_{T_{i,J}^{wx}} (u_{i,J} - u_{i-1,J}). \tag{A5}
$$

The second term evaluates as:

$$
\int_{\Delta V} \frac{\partial}{\partial y} \frac{1}{2}\left[ l_u(x,y)^2 \left| \frac{\partial u}{\partial y} \right| \left( \frac{\partial u}{\partial y} + \frac{\partial v}{\partial x} \right) \right] \, \mathrm{d}V = \frac{1}{2}\left[ l_u(x,y)^2 \left| \frac{\partial u}{\partial y} \right| \left( \frac{\partial u}{\partial y} + \frac{\partial v}{\partial x} \right) \right]_n \Delta x - \frac{1}{2}\left[ l_u(x,y)^2 \left| \frac{\partial u}{\partial y} \right| \left( \frac{\partial u}{\partial y} + \frac{\partial v}{\partial x} \right) \right]_s \Delta x.
$$

Here we have:

$$
\left.\frac{\partial u}{\partial y}\right|_n = \frac{u_{i,J+1} - u_{i,J}}{\Delta y_{J,J+1}}, \qquad \left.\frac{\partial v}{\partial x}\right|_n = \frac{v_{I,j+1} - v_{I-1,j+1}}{\Delta x_{I-1,I}}, \qquad \left.\frac{\partial u}{\partial y}\right|_s = \frac{u_{i,J} - u_{i,J-1}}{\Delta y_{J-1,J}}, \qquad \left.\frac{\partial v}{\partial x}\right|_s = \frac{v_{I,j} - v_{I-1,j}}{\Delta x_{I-1,I}},
$$

and $\Delta x = \Delta x_{I-1,I}$. Substituting yields:

$$
\int_{\Delta V} \frac{\partial}{\partial y} \frac{1}{2}\left[ l_u(x,y)^2 \left| \frac{\partial u}{\partial y} \right| \left( \frac{\partial u}{\partial y} + \frac{\partial v}{\partial x} \right) \right] \, \mathrm{d}V = \frac{1}{2}\left[ l_u(x_i,y_{j+1})^2 \left| \frac{u_{i,J+1} - u_{i,J}}{\Delta y_{J,J+1}} \right| \left( \frac{u_{i,J+1} - u_{i,J}}{\Delta y_{J,J+1}} + \frac{v_{I,j+1} - v_{I-1,j+1}}{\Delta x_{I-1,I}} \right) \right] \Delta x_{I-1,I} \ldots
$$
$$
- \frac{1}{2}\left[ l_u(x_i,y_j)^2 \left| \frac{u_{i,J} - u_{i,J-1}}{\Delta y_{J-1,J}} \right| \left( \frac{u_{i,J} - u_{i,J-1}}{\Delta y_{J-1,J}} + \frac{v_{I,j} - v_{I-1,j}}{\Delta x_{I-1,I}} \right) \right] \Delta x_{I-1,I},
$$





which can be rearranged to:

$$\int\limits_{\Delta V} \frac{\partial}{\partial y} \frac{1}{2}\left[l_u(x,y)^2 \left|\frac{\partial u}{\partial y}\right|\left(\frac{\partial u}{\partial y}+\frac{\partial v}{\partial x}\right)\right] \mathrm{d}V = \underbrace{\frac{1}{2}l_u(x_i,y_{j+1})^2 \left|\frac{(u_{i,J+1}-u_{i,J})\Delta x_{I-1,I}}{\Delta y^2_{J,J+1}}\right|}_{T^{nx}_{i,J}}(u_{i,J+1}-u_{i,J})\ldots$$

$$+\underbrace{\frac{1}{2}l_u(x_i,y_{j+1})^2 \left|\frac{(u_{i,J+1}-u_{i,J})}{\Delta y_{J,J+1}}\right|}_{T^{newx}_{i,J}}(v_{I,j+1}-v_{I-1,j+1})\ldots$$

$$-\underbrace{\frac{1}{2}l_u(x_i,y_j)^2 \left|\frac{(u_{i,J}-u_{i,J-1})\Delta x_{I-1,I}}{\Delta y^2_{J-1,J}}\right|}_{T^{sx}_{i,J}}(u_{i,J}-u_{i,J-1})\ldots$$

$$-\underbrace{\frac{1}{2}l_u(x_i,y_j)^2 \left|\frac{(u_{i,J}-u_{i,J-1})}{\Delta y_{J-1,J}}\right|}_{T^{sewx}_{i,J}}(v_{I,j}-v_{I-1,j}). \tag{A6}$$

Summarizing the above:

$$\frac{\partial}{\partial x}\left[l_u(x,y)^2 \left|\frac{\partial u}{\partial y}\right|\frac{\partial u}{\partial x}\right] + \frac{\partial}{\partial y}\frac{1}{2}\left[l_u(x,y)^2 \left|\frac{\partial u}{\partial y}\right|\left(\frac{\partial u}{\partial y}+\frac{\partial v}{\partial x}\right)\right] = \ldots$$

$$T^{ex}_{i,J}u_{i+1,J} + T^{wx}_{i,J}u_{i-1,J} + T^{nx}_{i,J}u_{i,J+1} + T^{sx}_{i,J}u_{i,J-1} + T^{px}_{i,J}u_{i,J} + T^{newx}_{i,J}(v_{I,j+1}-v_{I-1,j+1}) + T^{sewx}_{i,J}(v_{I-1,j}-v_{I,j}),$$

with $T^{px}_{i,J} = T^{ex}_{i,J} + T^{wx}_{i,J} + T^{nx}_{i,J} + T^{sx}_{i,J}$. The coefficients $T^{\bullet}_{i,J}$ will be computed for time $k$ while the flow components will be evaluated for time $k+1$.

**y-momentum equation**

Considering the y-momentum equation, the first term evaluates as: At last we derive, also for the y-momentum equation:

$$\int\limits_{\Delta V} \frac{\partial}{\partial y}\left[l_u(x,y)^2 \left|\frac{\partial u}{\partial y}\right|\frac{\partial v}{\partial y}\right] \mathrm{d}V = \left[l_u(x,y)^2 \left|\frac{\partial u}{\partial y}\right|\frac{\partial v}{\partial y}\right]_n \Delta x - \left[l_u(x,y)^2 \left|\frac{\partial u}{\partial y}\right|\frac{\partial v}{\partial y}\right]_s \Delta x.$$

Here we have:

$$\left.\frac{\partial u}{\partial y}\right|_n = \frac{u_{i+1,J}-u_{i+1,J-1}}{\Delta y_{J-1,J}}, \qquad \left.\frac{\partial v}{\partial y}\right|_n = \frac{v_{I,j+1}-v_{I,j}}{\delta y_{j,j+1}}, \qquad \left.\frac{\partial u}{\partial y}\right|_s = \frac{u_{i,J}-u_{i,J-1}}{\Delta y_{J-1,J}}, \qquad \left.\frac{\partial v}{\partial y}\right|_s = \frac{v_{I,j}-v_{I,j-1}}{\delta y_{j-1,j}},$$

and $\Delta x = \delta x_{i,i+1}$. Substituting these expressions yields:

$$\int\limits_{\Delta V} \frac{\partial}{\partial y}\left[l_u(x,y)^2 \left|\frac{\partial u}{\partial y}\right|\frac{\partial v}{\partial y}\right] \mathrm{d}V = \underbrace{l_u(x_I,y_J)^2 \left|\frac{(u_{i+1,J}-u_{i+1,J-1})\delta x_{i,i+1}}{\Delta y_{J-1,J}\delta y_{j,j+1}}\right|}_{T^{ny}_{I,j}}(v_{I,j+1}-v_{I,j})\ldots$$

$$-\underbrace{l_u(x_I,y_{J-1})^2 \left|\frac{(u_{i,J}-u_{i,J-1})\delta x_{i,i+1}}{\Delta y_{J-1,J}\delta y_{j-1,j}}\right|}_{T^{sy}_{I,j}}(v_{I,j}-v_{I,j-1}). \tag{A7}$$



The second term evaluates as:

$$\int_{\Delta V} \frac{\partial}{\partial x} \frac{1}{2} \left[ l_u(x,y)^2 \left| \frac{\partial u}{\partial y} \right| \left( \frac{\partial u}{\partial y} + \frac{\partial v}{\partial x} \right) \right] \mathrm{d}V = \frac{1}{2} \left[ l_u(x,y)^2 \left| \frac{\partial u}{\partial y} \right| \left( \frac{\partial u}{\partial y} + \frac{\partial v}{\partial x} \right) \right]_e \Delta y - \frac{1}{2} \left[ l_u(x,y)^2 \left| \frac{\partial u}{\partial y} \right| \left( \frac{\partial u}{\partial y} + \frac{\partial v}{\partial x} \right) \right]_w \Delta y.$$

Here we have:

$$\left. \frac{\partial u}{\partial y} \right|_e = \frac{u_{i+1,J} - u_{i+1,J-1}}{\Delta y_{J-1,J}}, \qquad \left. \frac{\partial v}{\partial x} \right|_e = \frac{v_{I+1,j} - v_{I,j}}{\Delta x_{I,I+1}}, \qquad \left. \frac{\partial u}{\partial y} \right|_w = \frac{u_{i,J} - u_{i,J-1}}{\Delta y_{J-1,J}}, \qquad \left. \frac{\partial v}{\partial x} \right|_w = \frac{v_{I,j} - v_{I-1,j}}{\Delta x_{I-1,I}},$$

and $\Delta y = \Delta y_{J-1,J}$. Substituting these expressions yields:

$$\int_{\Delta V} \frac{\partial}{\partial x} \frac{1}{2} \left[ l_u(x,y)^2 \left| \frac{\partial u}{\partial y} \right| \left( \frac{\partial u}{\partial y} + \frac{\partial v}{\partial x} \right) \right] \mathrm{d}V = \underbrace{\frac{1}{2} l_u(x_i, y_j)^2 \left| \frac{u_{i+1,J} - u_{i+1,J-1}}{\Delta y_{J-1,J}} \right| (u_{i+1,J} - u_{i+1,J-1})}_{T_{I,J}^{ensy}} \ldots$$

$$+ \underbrace{\frac{1}{2} l_u(x_i, y_j)^2 \left| \frac{u_{i+1,J} - u_{i+1,J-1}}{\Delta x_{I,I+1}} \right| (v_{I+1,j} - v_{I,j})}_{T_{I,j}^{ey}} \ldots$$

$$- \underbrace{\frac{1}{2} l_u(x_{i+1}, y_j)^2 \left| \frac{u_{i,J} - u_{i,J-1}}{\Delta y_{J-1,J}} \right| (u_{i,J} - u_{i,J-1})}_{T_{I,J}^{wnsy}} \ldots$$

$$- \underbrace{\frac{1}{2} l_u(x_{i+1}, y_j)^2 \left| \frac{u_{i,J} - u_{i,J-1}}{\Delta x_{I-1,I}} \right| (v_{I,j} - v_{I-1,j})}_{T_{I,J}^{wy}}. \tag{A8}$$

Summarizing the above:

$$\int_{\Delta V} \frac{\partial}{\partial y} \left[ l_u(x,y)^2 \left| \frac{\partial u}{\partial y} \right| \frac{\partial v}{\partial y} \right] \mathrm{d}V = \left[ l_u(x,y)^2 \left| \frac{\partial u}{\partial y} \right| \frac{\partial v}{\partial y} \right]_n \Delta x - \left[ l_u(x,y)^2 \left| \frac{\partial u}{\partial y} \right| \frac{\partial v}{\partial y} \right]_s \Delta x = \ldots$$

$$T_{I,j}^{ey} v_{I+1,j} + T_{I,j}^{wy} v_{I-1,j} + T_{I,j}^{ny} v_{I,j+1} + T_{I,j}^{sy} v_{I,j-1} + T_{I,j}^{py} v_{I,j} + T_{I,j}^{ensy}(u_{i+1,J} - u_{i+1,J-1}) + T_{i,J}^{wnsy}(u_{i,J} - u_{i,J-1}),$$

with $T_{I,j}^{py} = T_{I,j}^{ey} + T_{I,j}^{wy} + T_{I,j}^{ny} + T_{I,j}^{sy}$. The coefficients $T_{I,j}^{\bullet}$ will be computed for time $k$ while the flow components will be evaluated for time $k+1$.

## A4   Discretizing the forcing term

$$\int_{\Delta V} \frac{1}{2} \rho C_T' \left[ U \cos(\gamma) \right]^2 \begin{pmatrix} \cos(\gamma + \varphi) \\ \sin(\gamma + \varphi) \end{pmatrix} \mathrm{d}V = \frac{1}{2} \rho C_T' \left[ U \cos(\gamma) \right]^2 \begin{pmatrix} \cos(\gamma + \varphi) \\ \sin(\gamma + \varphi) \end{pmatrix} \Delta V$$

## A5   Discretizing the unsteady term

Evaluate:

$$\int_{\Delta V} \begin{pmatrix} \frac{\partial u}{\partial t} \\ \frac{\partial v}{\partial t} \end{pmatrix} \mathrm{d}V = \begin{pmatrix} \frac{\partial u}{\partial t} \\ \frac{\partial v}{\partial t} \end{pmatrix} \Delta V$$



Temporal discretization yields: $\frac{u_{k+1}-u_k}{\Delta t}$ and $\frac{v_{k+1}-v_k}{\Delta t}$ and we define:

$$a_0^{px} = \frac{\Delta V}{\Delta t} \qquad \text{and} \qquad a_0^{py} = \frac{\Delta V}{\Delta t}$$

### A6   Discretizing the Continuity equation

$$0 = \int\limits_{\Delta V} \frac{\partial u}{\partial x} + 2\frac{\partial v}{\partial y} \, \mathrm{d}V$$

$$= (u_{i+1,J} - u_{i,J})\,\delta y_{j,j+1} + 2\,(v_{I,j+1} - v_{I,j})\,\delta x_{i,i+1}.$$

All the coefficients derived above are given in Table 5.



**Table 5.** Fully discretized Navier-Stokes equations and all its coefficients.

x-momentum equation:

$$a_{i,J}^{px} u_{i,J} = \begin{pmatrix} a_{i,J}^{nx} & a_{i,J}^{sx} & a_{i,J}^{wx} & a_{i,J}^{ex} \end{pmatrix} \begin{pmatrix} u_{i,J+1} & u_{i,J-1} & u_{i-1,J} & u_{i+1,J} \end{pmatrix}^T - \delta y_{j,j+1}(p_{I,J} - p_{I-1,J}) + f_{i,J}^x + \ldots$$
$$+ \begin{pmatrix} a_{i,J}^{nwx} & a_{i,J}^{swx} & a_{i,J}^{nex} & a_{i,J}^{sex} \end{pmatrix} \begin{pmatrix} v_{I-1,j+1} & v_{I-1,j} & v_{I,j+1} & v_{I,J} \end{pmatrix}^T$$

y-momentum equation:

$$a_{I,j}^{py} v_{I,j} = \begin{pmatrix} a_{I,j}^{ny} & a_{I,j}^{sy} & a_{I,j}^{wy} & a_{I,j}^{ey} \end{pmatrix} \begin{pmatrix} v_{I,j+1} & v_{I,j-1} & v_{I-1,j} & v_{I+1,j} \end{pmatrix}^T - \delta x_{i,i+1}(p_{I,J} - p_{I,J-1}) + f_{I,j}^y + \ldots$$
$$+ \begin{pmatrix} a_{i,J}^{nwy} & a_{i,J}^{swy} & a_{i,J}^{ney} & a_{i,J}^{sey} \end{pmatrix} \begin{pmatrix} u_{i,J} & u_{i,J-1} & u_{i+1,J} & u_{i+1,J-1} \end{pmatrix}^T$$

continuity equation:

$$0 = \delta y_{j,j+1}(u_{i+1,J} - u_{i,J}) + 2\delta x_{i,i+1}(v_{I,j+1} - v_{I,j}),$$

$$a_{i,J}^{ex} = \max\left[-F_{i,J}^{ex}, 0\right] + T_{i,J}^{ex}, \qquad a_{i,J}^{wx} = \max\left[F_{i,J}^{wx}, 0\right] + T_{i,J}^{wx}, \qquad a_{i,J}^{nx} = \max\left[-F_{i,J}^{nx}, 0\right] + T_{i,J}^{nx}, \qquad a_{i,J}^{sx} = \max\left[F_{i,J}^{sx}, 0\right] + T_{i,J}^{sx},$$
$$a_{I,j}^{ey} = \max\left[-F_{I,j}^{ey}, 0\right] + T_{I,j}^{ey}, \qquad a_{I,j}^{wy} = \max\left[F_{I,j}^{wy}, 0\right] + T_{I,j}^{wy}, \qquad a_{I,j}^{ny} = \max\left[-F_{I,j}^{ny}, 0\right] + T_{I,j}^{ny}, \qquad a_{I,j}^{sy} = \max\left[F_{I,j}^{sy}, 0\right] + T_{I,j}^{sy},$$

$$a_{i,J}^{nex} = T_{i,J}^{newx}, \qquad a_{i,J}^{nwx} = T_{i,J}^{newx}, \qquad a_{i,J}^{sex} = T_{i,J}^{sewx}, \qquad a_{i,J}^{swx} = T_{i,J}^{sewx},$$
$$a_{I,j}^{ney} = T_{I,j}^{ensy}, \qquad a_{I,j}^{nwy} = T_{I,j}^{wnsy}, \qquad a_{I,j}^{sey} = T_{I,j}^{ensy}, \qquad a_{I,j}^{swy} = T_{I,j}^{wnsy},$$

$$a_{i,J}^{px} = a_{i,J}^{nx} + a_{i,J}^{ex} + a_{i,J}^{sx} + a_{i,J}^{wx} + F_{i,J}^{nx} + F_{i,J}^{ex} - F_{i,J}^{sx} - F_{i,J}^{wx} + T_{i,J}^{px} + a_0^{px},$$
$$a_{I,j}^{py} = a_{I,j}^{ny} + a_{I,j}^{ey} + a_{I,j}^{sy} + a_{I,j}^{wy} + F_{I,j}^{ny} + F_{I,j}^{ey} - F_{I,j}^{sy} - F_{I,j}^{wy} + T_{I,j}^{py} + a_0^{py},$$

in which:

$$F_{i,J}^{ex} = \tfrac{1}{2}\rho(u_{i+1,J} + u_{i,J})\delta y_{j,j+1}, \qquad\qquad F_{i,J}^{wx} = \tfrac{1}{2}\rho(u_{i,J} + u_{i-1,J})\delta y_{j,j+1},$$
$$F_{i,J}^{nx} = \tfrac{1}{2}\rho(v_{I,j+1} + v_{I-1,j+1})\Delta x_{I-1,I}, \qquad\qquad F_{i,J}^{sx} = \tfrac{1}{2}\rho(v_{I,j} + v_{I-1,j})\Delta x_{I-1,I},$$
$$F_{I,j}^{ey} = \tfrac{1}{2}\rho(u_{i+1,J} + u_{i+1,J-1})\Delta y_{J-1,J}, \qquad\qquad F_{I,j}^{wy} = \tfrac{1}{2}\rho(u_{i,J} + u_{i,J-1})\Delta y_{J-1,J},$$
$$F_{i,J}^{ny} = \tfrac{1}{2}\rho(v_{I,j+1} + v_{I-1,j+1})\Delta x_{I-1,I}, \qquad\qquad F_{i,J}^{sy} = \tfrac{1}{2}\rho(v_{I,j} + v_{I-1,j})\Delta x_{I-1,I},$$

$$a_0^{px} = \frac{\Delta x_{I-1,I}\delta y_{j,j+1}}{\Delta t}, \qquad\qquad a_0^{py} = \frac{\Delta y_{J-1,J}\delta x_{i,i+1}}{\Delta t},$$
$$\Delta x_{I-1,I} = x_I - x_{I-1}, \qquad\qquad \Delta y_{J-1,J} = y_J - y_{J-1},$$

$$T_{i,J}^{px} = T_{i,J}^{ex} + T_{i,J}^{wx} + T_{i,J}^{sx} + T_{i,J}^{nx}, \qquad \text{with } T_{i,J}^{\bullet} \text{ given in Eq. (A5) and Eq. (A6),}$$
$$T_{I,j}^{py} = T_{I,j}^{ey} + T_{I,j}^{wy} + T_{I,j}^{sy} + T_{I,j}^{ny}, \qquad \text{with } T_{I,j}^{\bullet} \text{ given Eq. (A7) and Eq. (A8),}$$

and:

$$f_{i,J}^x = \tfrac{1}{2}\delta y_{j,j+1}\rho C_T'[U_k \cos(\gamma_k)]^2 \cos(\gamma_k + \varphi_k), \qquad f_{I,j}^y = \tfrac{1}{2}\delta y_{J-1,J}\rho C_T'[U_k \cos(\gamma_k)]^2 \sin(\gamma_k + \varphi_k), \qquad U_k = \sqrt{u_{i,J}^2 + v_{I,j}^2}\cos(\gamma_k)$$





## Appendix B: PALM case study

In this appendix, a resolved flow field for arbitrary chosen time step is depicted for the PALM case study presented in §3.2.1.
Table 6 gives a summary of the PALM simulation set–up.

**Table 6.** Summary of the simulation set–up.

| | | | |
|---|---|---|---|
| Domain size $L_x \times L_y \times L_z$ | $19.2 \times 2.56 \times 1.28$ [km$^2$] | Turbine dimensions | $D =126$ [m], $z_h =90$ [m] |
| Grid size $N_x \times N_y \times N_z$ | $1920 \times 256 \times 1280$ | Turbine arrangement | $2 \times 1$ |
| Cell size $\Delta x \times \Delta y$ | $10 \times 10 \times 15$ [m$^2$] | Turbine spacing | $6D$ |
| Sample period $\Delta t$ | 1 [s] | Atmospheric conditions | $u_b = 8, v_b = 0, w_b = 0$ [m/s], $\rho = 1.2$ [kg/m$^3$] |
| Simulation time $t$ | 1750 [s] | Inflow | uniform |

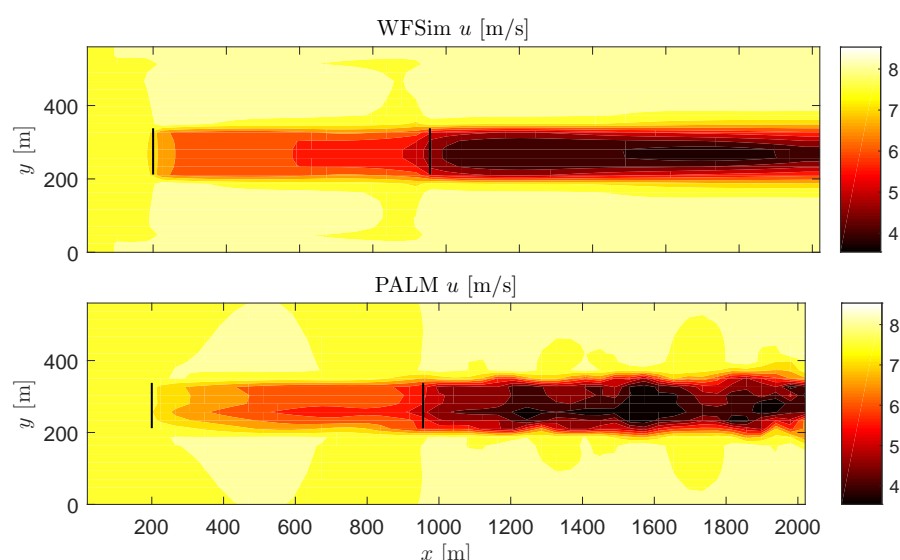

**Figure 15.** Flow field obtained with PALM (below) and WFSim at $t = 750$ [s]. The black lines indicate the turbines.





## Appendix C: SOWFA case study

In this appendix, a resolved flow field for arbitrary chosen time step is depicted for the SOWFA case study presented in §3.2.2. The SOWFA data set presented in van Wingerden et al. (2017) is utilized.

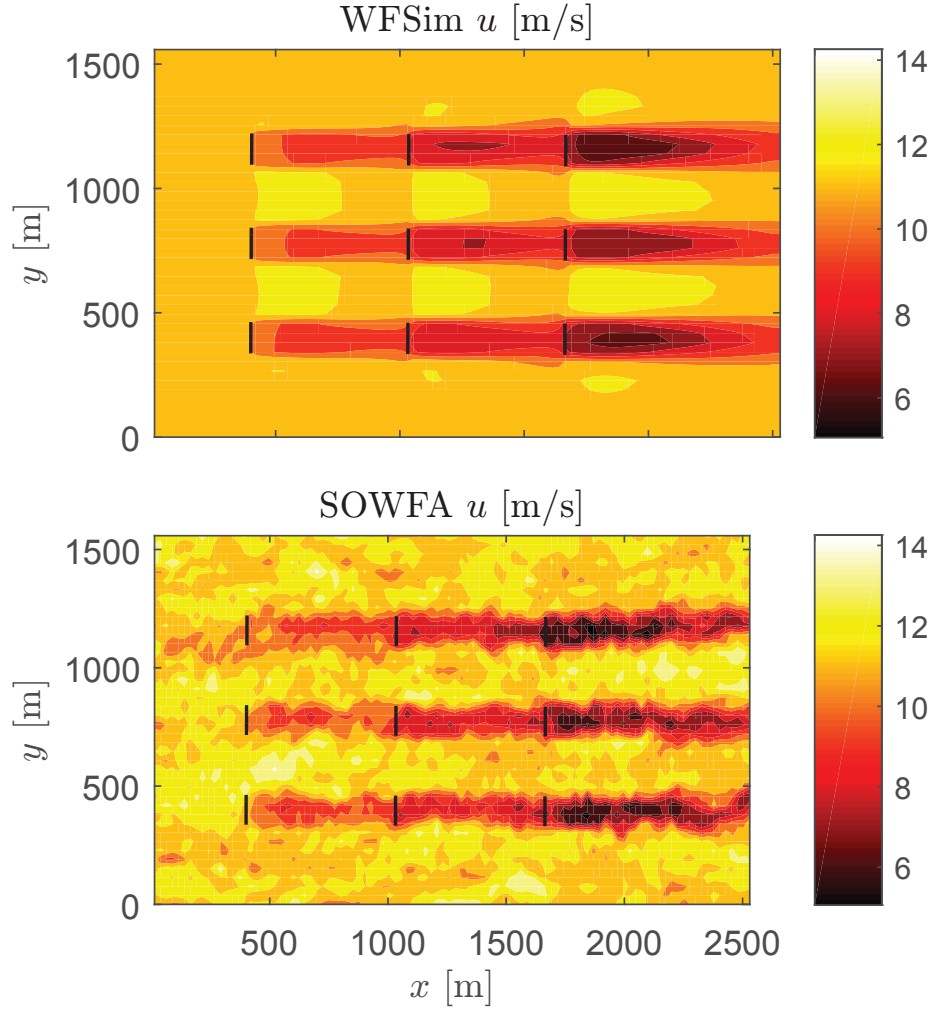

**Figure 16.** Flow field obtained with SOWFA (below) and WFSim at $t = 250$ [s]. The black lines indicate the turbines.





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
