# Peer review of "A control-oriented dynamic wind farm model: WFSim"

_Wind Energy Science, 2017_

## Referee Comment (RC1) · Anonymous Referee #1 · 6 Dec 2017

Dear authors,

Thank you for this contribution. The paper is very thoroughly written and complete. I agree that the model fills an important functional requirement of dynamic model for use in control with a computational efficiency suitable for online applications. Further, the innovations for efficiently incorporating important 3D effects is important. The paper is of high quality and should be accepted.

My primary comment is on intended application. The author's don't mention power-increasing wind farm control. Is that deliberate, in that is model is being designed specifically for electrical grid service provision?

A second question is on the analysis presented, do I understand correctly that for the

PALM and SOWFA comparisons, identical Ct time series are played through the turbine models, effectively open loop? Is there no online estimation being applied? And if not, is the assumption that if such estimation, made possible by the model structure, remove any remaining error?

---

## Short Comment (SC1) · 12 Dec 2017

Dear Referee #1,

Thanks for reviewing our work and the comments/questions. Below I provide answers.

» The author's don't mention powerincreasing wind farm control. Is that deliberate, in that is model is being designed specifically for electrical grid service provision?

Answer:

The presented wind farm model can be used in/for controllers providing grid facilities, but also controllers providing power maximization. In fact, in both:

Vali, M., van Wingerden, J. W., Boersma, S., Petrovic, V., and Kühn, M.: A predictive control framework for optimal energy extraction of wind farms, Journal of Physics: Conference Series, 2016.

Vali, M., Petrovic, V., Boersma, S., van Wingerden, J. W., and Kühn, M.: Adjoint-based model predictive control of wind farms: Beyond the quasi steady-state power maximization, International Federation of Automatic Control, 2017.

the objective is to maximize the power production of the farm.

» Do I understand correctly that for the PALM and SOWFA comparisons, identical Ct time series are played through the turbine models, effectively open loop?

Answer:

The signals are not equivalent for the PALM and SOWFA case. The signals are constructed such that they contain power in multiple frequency regions ensuring that the wind farm will be excited in multiple regions. In this way, we can show that for a broad frequency range, the presented model can estimate LES data. But, the CT' series applied to PALM and WFSim are exactly equivalent. The CT' series applied to WFSim are not exactly equivalent as applied in SOWFA since the latter does not allow for such a control input. We used equation 24 to estimate CT' (this is applied in WFSim) from SOWFA data.

» Is there no online estimation being applied?

Answer:

There is no online estimation applied in this work.

» Is the assumption that if such estimation, made possible by the model structure, remove any remaining error?

Answer:

In the following work:

Doekemeijer, B. M., van Wingerden, J. W., Boersma, S., and Pao, L. Y.: Enhanced Kalman filtering for a 2D CFD NS wind farm flow model, 20 Wind turbine wake estimation and control using floridyn, a control-oriented dynamic wind plant model., 2016.

Doekemeijer, B. M., Boersma, S., van Wingerden, J. W., and Pao, L. Y.: Ensemble Kalman filtering for wind field estimation in wind farms, American Control Conference, 2017.

we illustrate that the model can be used for online estimation and we illustrate that the estimation of wind farm dynamics will improve using an estimator (Ensemble Kalman filter in this case).

However, I don't think that any remaining error can be removed using an estimator, but also think that this is not the objective. The question according to me is: can the wind farm model (including online estimator) sufficiently capture the wind farm dynamics such that it can be employed for controller design/application providing pre-specified performance? And we are trying to answer this question by using the presented wind farm model as a building block for the proposed closed-loop control framework (Figure.1 in the paper).

I hope that your questions are sufficiently answered. Follow up questions are more than welcome.

Best regards, Sjoerd Boersma

---

## Referee Comment (RC2) · Anonymous Referee #2 · 19 Dec 2017

Dear Authors,

your paper is interesting and well written. I recommend publication after addressing the following points.

(1) Page 2: The first paragraph ob the abstracts reads like part of an introduction. Please reformulate the abstract in a more functional manner. Specify what you mean by "validated with high fidelity data" and give a rough summary of the findings. Also, preferably use present or past tense instead of "will be" formulations.

(2) Pages 3/4: In the model summary, please fill the gap between LES models and engineering models with 3D RANS references.

(3) Page 5: Indicate what you mean by "high fidelity simulation data" and the corre-

sponding site/wind farm characterisics (number of turbines, turbine types, ...)

(4) Page 7, first paragraph: Please clarify if the axial symmetry is assumed or not, in other words elaborate on the function tilde_v3(x,y,z). How to imagine the 2D model embedded into 3D space?

(5) Page 8: Please elaborate on the physics behind or the purpose of the function G.

(6) Page 9: The mixing length is for example unequal zero within the shaded area of turbine n in the central region of Fig. 3 (beginning of the curved arrow), i.e., at the region of the ramp (to which that arrow is pointing). Why should the ramp have that jump to zero at $x\_n' = d$? In other words, why is d smaller infinity? What does that mean physically? What is the physical meaning of the beginning of the ramp at $x\_n' = d'$? Please make the model a little more plausible to the reader. What is the physical implication of zero mixing length everywhere else?

(7) Page 9: The vector s is undefined.

(8) Page 14: What is meant by "a regular notebook"? Which programming language was used for the implementation?

(9) Page 14: What is the meaning of "relatively small"?

(10) Did you check mesh convergence for the presented results? Please add a comment or a graph.

(11) Please quantify explicitly the calculation times for all results. How does it relate to the response time of the controller and the chosen time step of the simulation?

---

## Author Comment (AC2) · 5 Jan 2018

Dear Referee #2,

We thank you for your feedback. Please find our responses to your questions below.

(1) Page 2: The first paragraph of the abstracts reads like part of an introduction. Please reformulate the abstract in a more functional manner. Specify what you mean by "validated with high fidelity data" and give a rough summary of the findings. Also, preferably use present or past tense instead of "will be" formulations.

Answer: Thank you for this valuable comment. The abstract will be rewritten and we will specify more precisely what we did in the paper.

[Figure]

(2) Pages 3/4: In the model summary, please fill the gap between LES models and engineering models with 3D RANS references.

Answer: The following:

Crespo, A., Hernandez, J., Fraga, E., and Andreu, C.: Experimental validation of the UPM computer code to calculate wind turbine wakes and comparison with other models, Journal of Wind Engineering and Industrial Aerodynamics, 1988.

and

Özdemir, H., Versteeg, M. C., and Brand, A. J.: Improvements in ECN wake model, ICOWES conference, 2013.

are both referring to a 3D RANS wind farm model. We will mention this more specific.

In addition, the following references will be included:

M. Avila, A. Folch, G. Houzeaux, B. Eguzkitza, L. Prieto, D. Cabezón, A Parallel CFD Model for Wind Farms, In Procedia Computer Science, Volume 18, 2013, Pages 2157-2166.

van der Laan, M. Paul & Sørensen, Niels & Réthoré, Pierre-Elouan & Mann, Jakob & Kelly, Mark & Troldborg, Niels. (2014). The k-$\varepsilon$-fP model applied to double wind turbine wakes using different actuator disk force methods. Wind Energy.

We would like to hear if then still a reference is missing.

(3) Page 5: Indicate what you mean by "high fidelity simulation data" and the corresponding site/wind farm characteristics (number of turbines, turbine types, ...)

Answer: With high-fidelity simulation data we mean flow velocities in the x- and y-direction at hub-height and turbine power signals computed with a LES based wind farm model. In the revised version of the paper we will specify this in the introduction and also the number of turbines and their specifications.

(4) Page 7, first paragraph: Please clarify if the axial symmetry is assumed or not, in other words elaborate on the function tilde_v3(x,y,z). How to imagine the 2D model embedded into 3D space?

Answer: We do not assume axial symmetry, and in particular we do not define a symmetry axis (which would be necessary to start with when introducing axial symmetry). Nevertheless, our choice (w~0 and dw/dz ~ dv/dy) is inspired by conditions required for axial symmetry. If a single turbine is considered, and we look at a streamline along the turbine axis, axial symmetry implies indeed w~0 and dw/dz ~ dv/dy, but requires further conditions on dv/dz and dw/dy (which we do not impose in our model). Away from the turbine axis, these conditions are not consistent anymore with axial symmetry, nor are they for a full wind farm case with multiple turbines. They rather imply equal divergence/convergence of streamlines in y and z direction. We do not expect this assumption to be accurate everywhere, but we presume it to be good enough to resolve the lack of relaxation of purely 2D models. If necessary, a more general form (w~0 and dw/dz ~ c dv/dy), with c a tuning parameter (e.g. obtained through state estimation) could be considered, but results in the current work indicate that this may not be necessary. We will incorporate these issues better in the revision, using above discussion.

(5) Page 8: Please elaborate on the physics behind or the purpose of the function G.

Answer: The function G is a smoothing function that ensures a mixing length parameter that is differentiable. Also, from a physical point of view, it is more realistic to smoothly change the turbulence characteristics instead of abruptly.

(6) Page 9: The mixing length is for example unequal zero within the shaded area of turbine n in the central region of Fig. 3 (beginning of the curved arrow), i.e., at the region of the ramp (to which that arrow is pointing). Why should the ramp have that jump to zero at $x_n' = d$? In other words, why is d smaller infinity? What does that mean physically? What is the physical meaning of the beginning of the ramp at $x_n' =$

d'? Please make the model a little more plausible to the reader. What is the physical implication of zero mixing length everywhere else?

Answer: The jump at zero is smoothened out by the function G.

At this point we would like to stress that we developed an engineering model and some of the parameter are introduced for tuning purposes. In general we could say that the turbulence model is parameterized by $l\_s,d$, and $d'$. It allows us to regulate in which area in the wind farm model we would like to have more or less wake recovery. Increasing values of $l\_u^n$ results in more wake recovery. Thus, if we allow d to be infinite, the wake recovery would also increase when the wake passes a downwind turbine. This is from a physical point of view not realistic since much more physics is happening when wind passes a turbine.

By setting the mixing length parameter to zero downwind of a turbine (before a downwind turbine) we could say that we reset the wake recovery again. This parametrization is also not based on pure physical reasoning, but captures a change in wake recovery after flowing through a rotor. We are using a linearly increasing mixing length $l\_u^n$ behind the rotor following the results presented in:

Iungo, G. V., Viola, F., Ciri,U., Rotea,M. A., and Leo: Data-driven RANS for simulations of large wind farms, Journal of Physics: Conference Series, 2015.

Overall we think that it is good to mention that the included turbulence model is a simplified mixing length model found heuristically using and adapting information from the reference above. We will state this more clearly in Section 2.1.

(7) Page 9: The vector s is undefined.

Answer: $s = [x,y]$, i.e., it indicates a position in the farm in the x,y coordinate frame. We will also indicate this in the text.

(8) Page 14: What is meant by "a regular notebook"? Which programming language was used for the implementation?

Answer: The programming language is MATLAB and the notebook contains an Intel Core i7 2.7 GHz processor. We will explicitly mention this in the revised version, also that the simulations are done using a single core.

(9) Page 14: What is the meaning of "relatively small"?

Answer: Order of magnitude three smaller. Basically, elements in the off-diagonal matrices are of order $O(1)$ while the elements in the diagonal matrices are of the order $O(10^3)$. We will mention this more explicitly in the paper.

(10) Did you check mesh convergence for the presented results? Please add a comment or a graph.

Answer: We did not study mesh convergence in detail, but rather looked into whether LES data could be approximated with the presented simplified wind farm model. We will include a note regarding this subject in the paper, but are not planning to do such a study in this work.

(11) Please quantify explicitly the calculation times for all results. How does it relate to the response time of the controller and the chosen time step of the simulation? Answer: The two-turbine case presented in Section 3.2.1 takes 0.02 sec per time step and the nine-turbine case presented in Section 3.2.2 takes 0.1 sec per time step. We will explicitly write these CPU times in the corresponding paragraphs.

Assuming that the controller contains the presented model, it is difficult to say what the controller response time will be. This depends on how the presented model is used in the controller. When applying the model in a model predictive controller, the controller response time depends, e.g., on the prediction horizon. Also, the employed optimization procedure could demand for line search techniques or a backward simulation. In such a case, additional trajectories need to be simulated.

However, the objective of this work is to keep the CPU time of the model as low as possible, while still capturing the dominant wind farm dynamics relevant for control

purposes. The objective of the controller design should also be to keep the CPU times sufficiently low such that online control can be achieved.

The time step of the simulations could be chosen larger or smaller while not changing the CPU time per time step and not affecting stability (we employ an implicit discretisation method). However, we did not validate the WFSim simulation results in such cases. This could be an interesting study.

---

## Author Response (AR1)

*Referee #1*

A-----------------------------------
(1)
The author's don't mention power increasing wind farm control. Is that deliberate, in that is model is being designed specifically for electrical grid service provision?

(2)
The presented wind farm model can potentially be used in/for controllers providing grid facilities as demonstrated in

Vali, M., Petrovic, V., Boersma, S., van Wingerden, J. W., Pao, L.Y. and Kühn, M.: Model Predictive Active Power Control of Waked Wind Farms, American Control Conference, 2018 (under review).

, but also controllers providing power maximization. In fact, in both:

Vali, M., van Wingerden, J. W., Boersma, S., Petrovic, V., and Kühn, M.: A predictive control framework for optimal energy extraction of wind farms, Journal of Physics: Conference Series, 2016.

Vali, M., Petrovic, V., Boersma, S., van Wingerden, J. W., and Kühn, M.: Adjoint based model predictive control of wind farms: Beyond the quasi steady-state power maximization, International Federation of Automatic Control, 2017.

the objective is to maximize the power production of the farm. In the revised version of this paper, we will emphasise the fact that this model can potentially also be used for power optimization.

While the above results are promising, they are obtained with controllers using the same wind farm model (WFSim) as to which the found control signals are applied (called the simulation model). In other words, perfect system knowledge is assumed. Similarly, in

Munters, W. and Meyers, J.: An optimal control framework for dynamic induction control of wind farms and their interaction with the atmospheric boundary layer. *Phil. Trans. R. Soc. A* **375**: 20160100.

the authors illustrate the potential of power maximization using a LES based wind farm model as the simulation model and as model in the controller. While a LES based model is relatively accurate, it is also computationally complex and therefore not suitable for online control.

At the moment, we are investigating if a combination of the above results can give satisfying controller performance. More precisely, the model in the controller should be WFSim (due to its computational efficiency) and the simulation model should be a LES based wind farm model.

(3)
-

B-----------------------------------
(1)
Do I understand correctly that for the PALM and SOWFA comparisons, identical Ct time series are played through the turbine models, effectively open loop?

(2)
    a)   The control signals applied in the PALM and SOWFA case are not equivalent.

b) The CT' series applied to PALM and WFSim are exactly equivalent.
c) The CT' series applied to WFSim are not exactly equivalent as applied in SOWFA since the latter does not allow for such a control input. We used equation 24 to estimate CT' (this is applied in WFSim) from SOWFA data.

(3)
-

C------------------------------------
(1) Is there no online estimation being applied?

(2)
There is no online estimation applied in this work. However, the presented model can be used for online estimation (see citations in next answer).

(3)
-

D------------------------------------
(1)
Is the assumption that if such estimation, made possible by the model structure, remove any remaining error?

(2)
In the following work:

Doekemeijer, B. M., van Wingerden, J. W., Boersma, S., and Pao, L. Y.: Enhanced Kalman filtering for a 2D CFD NS wind farm flow model, Journal of Physics: Conference Series, 2016.

Doekemeijer, B. M., Boersma, S., van Wingerden, J. W., and Pao, L. Y.: Ensemble Kalman filtering for wind field estimation in wind farms, American Control Conference, 2017.

we illustrate that the model can be used for online estimation and we illustrate that the estimation of wind farm dynamics will be improved by using an estimator (Ensemble Kalman filter in this case).

The purpose of the WindFarmSimulator model is not to capture all the flow and turbine dynamics that LES models typically capture. Rather, the objective is to capture the dominant spatial and temporal dynamics inside the wind farm to allow reliable forecasting of each turbine's power generation in a time efficient manner. This in turn enables wind farm control algorithms to, e.g., consistently track a desired power reference signal by predicting the effect of turbine control policies on the surrounding wind turbines. In the bigger picture, we propose a closed-loop control solution in which the WFSim model is calibrated in real-time to model discrepancies and to the current atmospheric conditions inside the wind farm. This calibrated model is then used for forecasting and for determining a control policy. The proposed closed-loop control framework is displayed in Fig. 1 in the paper.

(3)
-

**Referee #2**

A-----------------------------------
(1)
Page 2: The first paragraph of the abstracts reads like part of an introduction.
Please reformulate the abstract in a more functional manner. Specify what you mean
by "validated with high fidelity data" and give a rough summary of the findings. Also,
preferably use present or past tense instead of "will be" formulations.

(2)
Thank you for this valuable comment. The abstract will be rewritten and we will specify more
precisely what we did in the paper.

(3)
The abstract is rewritten. Point 1 in the revised version.

B-----------------------------------
(1)
Pages 3/4: In the model summary, please fill the gap between LES models and
engineering models with 3D RANS references.

(2)
The following:

Crespo, A., Hernandez, J., Fraga, E., and Andreu, C.: Experimental validation of the UPM computer
code to calculate wind turbine wakes and comparison with other models, Journal of Wind
Engineering and Industrial Aerodynamics, 1988.

and

Özdemir, H., Versteeg, M. C., and Brand, A. J.: Improvements in ECN wake model, ICOWES
conference, 2013.

are both referring to a 3D RANS wind farm model. We will mention this more specific.

In addition, the following references will be included:

M. Avila, A. Folch, G. Houzeaux, B. Eguzkitza, L. Prieto, D. Cabezón, A Parallel CFD Model for Wind
Farms, In Procedia Computer Science, Volume 18, 2013, Pages 2157-2166.

van der Laan, M. Paul & Sørensen, Niels & Réthoré, Pierre-Elouan & Mann, Jakob & Kelly, Mark &
Troldborg, Niels. (2014). The k-ε-fP model applied to double wind turbine wakes using different
actuator disk force methods. Wind Energy.

(3)
References are added. Point 2 in the revised version.

C-----------------------------------
(1) Page 5: Indicate what you mean by "high fidelity simulation data" and the corresponding
site/wind farm characteristics (number of turbines, turbine types, ...)

(2)
With high-fidelity simulation data we mean flow velocities in the x- and y-direction at hub-height and turbine power signals computed with a LES based wind farm model.

(3)
In the revised version of the paper we specified this in the introduction and also the number of turbines and their specifications. Point 3 in the revised version.

D------------------------------------
(1)
Page 7, first paragraph: Please clarify if the axial symmetry is assumed or not, in other words elaborate on the function $tilde\_v3(x,y,z)$. How to imagine the 2D model embedded into 3D space?

(2)
We do not assume axial symmetry, and in particular we do not define a symmetry axis (which would be necessary to start with when introducing axial symmetry). Nevertheless, our choice (w≈0 and $dw/dz \approx dv/dy$) is inspired by conditions required for axial symmetry. If a single turbine is considered, and we look at a streamline along the turbine axis, axial symmetry implies indeed w≈0 and $dw/dz \approx dv/dy$, but requires further conditions on $dv/dz$ and $dw/dy$ (which we do not impose in our model). Away from the turbine axis, these conditions are not consistent anymore with axial symmetry, nor are they for a full wind farm case with multiple turbines. They rather imply equal divergence/convergence of streamlines in y and z direction. We do not expect this assumption to be accurate everywhere, but we presume it to be good enough to resolve the lack of relaxation of purely 2D models. If necessary, a more general form (w≈0 and $dw/dz \approx c\, dv/dy$), with c a tuning parameter (e.g. obtained through state estimation) could be considered, but results in the current work indicate that this may not be necessary. We will incorporate these issues better in the revision, using above discussion.

(3)
We incorporated these issues better in the revision, using above discussion. Point 4 in the revised version.

E------------------------------------
(1)
Page 8: Please elaborate on the physics behind or the purpose of the function G.

(2)
The function G is a smoothing function that ensures a mixing length parameter that is differentiable. Also, from a physical point of view, it is more realistic to smoothly change the turbulence characteristics instead of abruptly.

(3)
-

F------------------------------------
(1)
Page 9: The mixing length is for example unequal zero within the shaded area of turbine n in the central region of Fig. 3 (beginning of the curved arrow), i.e., at the region of the ramp (to which that arrow is pointing). Why should the ramp have that jump to zero at $x\_n' = d$? In other words, why is d smaller infinity? What does that

mean physically? What is the physical meaning of the beginning of the ramp at $x\_n' = d'$? Please make the model a little more plausible to the reader. What is the physical implication of zero mixing length everywhere else?

(2)
The jump at zero is smoothened out by the function G.

At this point we would like to stress that we developed an engineering model and some of the parameter are introduced for tuning purposes. In general we could say that the turbulence model is parameterized by $l\_s, d$, and $d'$. It allows us to regulate in which area in the wind farm model we would like to have more or less wake recovery. Increasing values of $l\_u^n$ results in more wake recovery. Thus, if we allow $d$ to be infinite, the wake recovery would also increase when the wake passes a downwind turbine. This is from a physical point of view not realistic since much more physics is happening when wind passes a turbine.

By setting the mixing length parameter to zero downwind of a turbine (before a downwind turbine) we could say that we reset the wake recovery again. This parametrization is also not based on pure physical reasoning, but captures a change in wake recovery after flowing through a rotor. We are using a linearly increasing mixing length $l\_u^n$ behind the rotor following the results presented in:

Iungo, G. V., Viola, F., Ciri,U., Rotea,M. A., and Leo: Data-driven RANS for simulations of large wind farms, Journal of Physics: Conference Series, 2015.

(3)
Overall we think that it is good to mention that the included turbulence model is a simplified mixing length model found heuristically using and adapting information from the reference above. We stated this more clearly in the revised version. Point 6 in the revised version.

G------------------------------------
(1)
Page 9: The vector s is undefined.

(2)
s = [x,y], i.e., it indicates a position in the farm in the x,y coordinate frame.

(3)
The vector s is defined in the revised version. Point 7 in the revised version.

H------------------------------------
(1)
Page 14: What is meant by "a regular notebook"? Which programming language was used for the implementation?

(2)
The programming language is MATLAB and the notebook contains an Intel Core i7 2.7 GHz processor.

(3)
We explicitly mention this in the revised version, also that the simulations are done using a single core. Point 8 in the revised version.

I-----------------------------------
(1)
Page 14: What is the meaning of "relatively small"?

(2)
Order of magnitude three smaller. Basically, elements in the off-diagonal matrices are of order $O(1)$ while the elements in the diagonal matrices are of the order $O(10^3)$.

(3)
We mention this more explicitly in the paper. Point 9 in the revised version.

J-----------------------------------
(1)
Did you check mesh convergence for the presented results? Please add a comment or a graph.

(2)
We did not study mesh convergence in detail, but rather looked into whether LES data could be approximated with the presented simplified wind farm model.

(3)
We included a note regarding this subject in the paper. Point 10 in the revised version.

K-----------------------------------
(1)
Please quantify explicitly the calculation times for all results. How does it relate to the response time of the controller and the chosen time step of the simulation?

(2)
The two-turbine case presented in Section 3.2.1 takes 0.02 sec per time step and the nine-turbine case presented in Section 3.2.2 takes 0.1 sec per time step.

Assuming that the controller contains the presented model, it is difficult to say what the controller response time will be. This depends on how the presented model is used in the controller. When applying the model in a model predictive controller, the controller response time depends, e.g., on the prediction horizon. Also, the employed optimization procedure could demand for line search techniques or a backward simulation. In such a case, additional trajectories need to be simulated.

However, the objective of this work is to keep the CPU time of the model as low as possible, while still capturing the dominant wind farm dynamics relevant for control purposes. The objective of the controller design should also be to keep the CPU times sufficiently low such that online control can be achieved.

The time step of the simulations could be chosen larger or smaller while not changing the CPU time per time step and not affecting stability (we employ an implicit discretisation method). However, we did not validate the WFSim simulation results in such cases. This could be an interesting study.

(3)
We explicitly wrote these CPU times in the corresponding paragraphs. Point 11 in the revised version.

[revised manuscript text omitted]